# Online Continuous Submodular Maximization: From Full-Information to Bandit Feedback

**Mingrui Zhang**[†]    **Lin Chen**[‡]    **Hamed Hassani**[♯]    **Amin Karbasi**[‡,♮]

[†] Department of Statistics and Data Science, Yale University
[‡] Department of Electrical Engineering, Yale University
[♯] Department of Electrical and Systems Engineering, University of Pennsylvania
[♮] Department of Computer Science, Yale University
{mingrui.zhang, lin.chen, amin.karbasi}@yale.edu  hassani@seas.upenn.edu

## Abstract

In this paper, we propose three online algorithms for submodular maximization. The first one, `Mono-Frank-Wolfe`, reduces the number of per-function gradient evaluations from $T^{1/2}$ [18] and $T^{3/2}$ [17] to 1, and achieves a $(1-1/e)$-regret bound of $O(T^{4/5})$. The second one, `Bandit-Frank-Wolfe`, is the first bandit algorithm for continuous DR-submodular maximization, which achieves a $(1-1/e)$-regret bound of $O(T^{8/9})$. Finally, we extend `Bandit-Frank-Wolfe` to a bandit algorithm for discrete submodular maximization, `Responsive-Frank-Wolfe`, which attains a $(1-1/e)$-regret bound of $O(T^{8/9})$ in the responsive bandit setting.

## 1   Introduction

Submodularity naturally arises in a variety of disciplines, and has numerous applications in machine learning, including data summarization [45], active and semi-supervised learning [26, 47], compressed sensing and structured sparsity [7], fairness in machine learning [8], mean-field inference in probabilistic models [10], and MAP inference in determinantal point processes (DPPs) [36].

We say that a set function $f : 2^\Omega \to \mathbb{R}_{\geq 0}$ defined on a finite ground set $\Omega$ is *submodular* if for every $A \subseteq B \subseteq \Omega$ and $x \in \Omega \setminus B$, we have $f(x|A) \geq f(x|B)$, where $f(x|A) \triangleq f(A \cup \{x\}) - f(A)$ is a discrete derivative [39]. Continuous DR-submodular functions are the continuous analogue. Let $F : \mathcal{X} \to \mathbb{R}_{\geq 0}$ be a differentiable function defined on a box $\mathcal{X} \triangleq \prod_{i=1}^d \mathcal{X}_i$, where each $\mathcal{X}_i$ is a closed interval of $\mathbb{R}_{\geq 0}$. We say that $F$ is *continuous DR-submodular* if for every $x, y \in \mathcal{X}$ that satisfy $x \leq y$ and every $i \in [d] \triangleq \{1, \dots, d\}$, we have $\frac{\partial F}{\partial x_i}(x) \geq \frac{\partial F}{\partial x_i}(y)$, where $x \leq y$ means $x_i \leq y_i, \forall i \in [d]$ [9].

In this paper, we focus on online and bandit maximization of submodular set functions and continuous DR-submodular functions. In contrast to offline optimization where the objective function is completely known beforehand, online optimization can be viewed as a two-player game between the player and the adversary in a sequential manner [50, 42, 28]. Let $\mathcal{F}$ be a family of real-valued functions. The player wants to maximize a sequence of functions $F_1, \dots, F_T \in \mathcal{F}$ subject to a constraint set $\mathcal{K}$. The player has no *a priori* knowledge of the functions, while the constraint set is known and we assume that it is a closed convex set in $\mathbb{R}^d$. The natural number $T$ is termed the *horizon* of the online optimization problem. At the $t$-th iteration, without the knowledge of $F_t$, the player has to select a point $x_t \in \mathcal{K}$. After the player commits to this choice, the adversary selects a function $F_t \in \mathcal{F}$. The player receives a reward $F_t(x_t)$, observes the function $F_t$ determined by the adversary, and proceeds to the next iteration. In the more challenging bandit setting, even the

function $F_t$ is unavailable to the player and the only observable information is the reward that the player receives [23, 3, 11].

The performance of the algorithm that the player uses to determine her choices $x_1, \ldots, x_T$ is quantified by the regret, which is the gap between her accumulated reward and the reward of the best single choice in hindsight. To be precise, the regret is defined by $\max_{x \in \mathcal{K}} \sum_{t=1}^{T} F_t(x) - \sum_{t=1}^{T} F_t(x_t)$. However, even in the offline scenario, it is shown that the maximization problem of a continuous DR-submodular function cannot be approximated within a factor of $(1 - 1/e + \epsilon)$ for any $\epsilon > 0$ in polynomial time, unless $RP = NP$ [9]. Therefore, we consider the $(1 - 1/e)$-regret [44, 34, 18]

$$\mathcal{R}_{1-1/e,T} \triangleq (1 - 1/e) \max_{x \in \mathcal{K}} \sum_{t=1}^{T} F_t(x) - \sum_{t=1}^{T} F_t(x_t).$$

For ease of notation, we write $\mathcal{R}_T$ for $\mathcal{R}_{1-1/e,T}$ throughout this paper.

In this paper, we study the following three problems:

- `OCSM`: the Online Continuous DR-Submodular Maximization problem,
- `BCSM`: the Bandit Continuous DR-Submodular Maximization problem, and
- `RBSM`: the Responsive Bandit Submodular Maximization problem.

We note that although special cases of bandit submodular maximization problem (`BSM`) were studied in [44, 27], the vanilla `BSM` problem is still open for general monotone submodular functions under a matroid constraint. In `BSM`, the objective functions $f_1, \ldots, f_T$ are submodular set functions defined on a common finite ground set $\Omega$ and subject to a common constraint $\mathcal{I}$. For each function $f_i$, the player has to select a subset $X_i \in \mathcal{I}$. Only after playing the subset $X_i$, the reward $f_i(X_i)$ is received and thereby observed.

If the value of the corresponding multilinear extension[1] $F$ can be estimated by the submodular set function $f$, we may expect to solve the vanilla `BSM` by invoking algorithms for continuous DR-submodular maximization. In this paper, however, we will show a hardness result that subject to some constraint $\mathcal{I}$, it is impossible to construct a one-point unbiased estimator of the multilinear extension $F$ based on the value of $f$, without knowing the information of $f$ in advance. This result motivates the study of a slightly relaxed setting termed the *Responsive Bandit Submodular Maximization* problem (`RBSM`). In `RBSM`, at round $i$, if $X_i \notin \mathcal{I}$, the player is still allowed to play $X_i$ and observe the function value $f_i(X_i)$, but gets zero reward out of it.

`OCSM` was studied in [18, 17], where $T^{1/2}$ exact gradient evaluations or $T^{3/2}$ stochastic gradient evaluations are required per iteration ($T$ is the horizon). Therefore, they cannot be extended to the bandit setting (`BCSM` and `RBSM`) where one single function evaluation per iteration is permitted. As a result, no known bandit algorithm attains a sublinear $(1 - 1/e)$-regret.

In this paper, we first propose `Mono-Frank-Wolfe` for `OCSM`, which requires one stochastic gradient per function and still attains a $(1 - 1/e)$-regret bound of $O(T^{4/5})$. This is significant as it reduces the number of per-function gradient evaluations from $T^{3/2}$ to 1. Furthermore, it provides a feasible avenue to solving `BCSM` and `RBSM`. We then propose `Bandit-Frank-Wolfe` and `Responsive-Frank-Wolfe` that attain a $(1 - 1/e)$-regret bound of $O(T^{8/9})$ for `BCSM` and `RBSM`, respectively. To the best of our knowledge, `Bandit-Frank-Wolfe` and `Responsive-Frank-Wolfe` are the first algorithms that attain a sublinear $(1 - 1/e)$-regret bound for `BCSM` and `RBSM`, respectively.

The performance of prior approaches and our proposed algorithms is summarized in Table 1. We also list further related works in Appendix A.

## 2 Preliminaries

**Monotonicity, Smoothness, and Directional Concavity Property** A submodular set function $f : 2^{\Omega} \to \mathbb{R}$ is called *monotone* if for any two sets $A \subseteq B \subseteq \Omega$ we have $f(A) \leq f(B)$.

For two vectors $x$ and $y$, we write $x \leq y$ if $x_i \leq y_i$ holds for every $i$. Let $F$ be a continuous DR-submodular function defined on $\mathcal{X}$. We say that $F$ is *monotone* if $F(x) \leq F(y)$ for every $x, y \in \mathcal{X}$ obeying $x \leq y$. Additionally, $F$ is called *L-smooth* if for every $x, y \in \mathcal{X}$ it holds that

Table 1: Comparison of previous and our proposed algorithms.

| Setting | Algorithm | Stochastic gradient | # of grad. evaluations | $(1-1/e)$-regret |
|---------|-----------|---------------------|------------------------|------------------|
| OCSM | Meta-FW [18] | No | $T^{1/2}$ | $O(\sqrt{T})$ |
|  | VR-FW [17] | Yes | $T^{3/2}$ | $O(\sqrt{T})$ |
|  | Mono-FW (**this work**) | Yes | 1 | $O(T^{4/5})$ |
| BCSM | Bandit-FW (**this work**) | - | - | $O(T^{8/9})$ |
| RBSM | Responsive-FW (**this work**) | - | - | $O(T^{8/9})$ |

$\|\nabla F(x) - \nabla F(y)\| \leq L\|x - y\|$. Throughout the paper, we use the notation $\|\cdot\|$ for the Euclidean norm. An important implication of continuous DR-submodularity is concavity along the non-negative directions [16, 9], *i.e.*, for all $x \leq y$, we have $F(y) \leq F(x) + \langle \nabla F(x), y - x \rangle$.

**Multilinear Extension**   Given a submodular set function $f : 2^\Omega \to \mathbb{R}_{\geq 0}$ defined on a finite ground set $\Omega$, its *multilinear extension* is a continuous DR-submodular function $F : [0,1]^{|\Omega|} \to \mathbb{R}_{\geq 0}$ defined by $F(x) = \sum_{S \subseteq \Omega} f(S) \Pi_{i \in S} x_i \Pi_{j \notin S}(1 - x_j)$, where $x_i$ is the $i$-th coordinate of $x$. Equivalently, for any vector $x \in [0,1]^{|\Omega|}$ we have $F(x) = \mathbb{E}_{S \sim x}[f(S)]$ where $S \sim x$ means that $S$ is a random subset of $\Omega$ such that every element $i \in \Omega$ is contained in $S$ independently with probability $x_i$.

**Geometric Notations**   The $d$-dimensional unit ball is denoted by $B^d$, and the $(d-1)$-dimensional unit sphere is denoted by $S^{d-1}$. Let $\mathcal{K}$ be a bounded set. We define its diameter $D = \sup_{x,y \in \mathcal{K}} \|x - y\|$ and radius $R = \sup_{x \in \mathcal{K}} \|x\|$. We say a set $\mathcal{K}$ has lower bound $\underline{u}$ if $\underline{u} \in \mathcal{K}$, and $\forall x \in \mathcal{K}, x \geq \underline{u}$.

## 3   One-shot Online Continuous DR-Submodular Maximization

In this section, we propose `Mono-Frank-Wolfe`, an online continuous DR-submodular maximization algorithm which only needs one gradient evaluation per function. This algorithm is the basis of the methods presented in the next section for the bandit setting. We also note that throughout this paper, $\nabla F$ denotes the exact gradient for $F$, while $\tilde{\nabla} F$ denotes the stochastic gradient.

We begin by reviewing the Frank-Wolfe (FW) [24, 33] method for maximizing monotone continuous DR-submodular functions in the offline setting [9], where we have one single objective function $F$. Assuming that we have access to the exact gradient $\nabla F$, the FW method is an iterative procedure that starts from the initial point $x^{(1)} = 0$, and at the $k$-th iteration, solves a linear optimization problem

$$v^{(k)} \leftarrow \arg\max_{v \in \mathcal{K}} \langle v, \nabla F(x^{(k)}) \rangle \tag{1}$$

which is used to update $x^{(k+1)} \leftarrow x^{(k)} + \eta_k v^{(k)}$, where $\eta_k$ is the step size.

We aim to extend the FW method to the online setting. Inspired by the FW update above, to get high rewards for each objective function $F_t$, we start from $x_t^{(1)} = 0$, update $x_t^{(k+1)} = x_t^{(k)} + \eta_k v_t^{(k)}$ for multiple iterations (let $K$ denote the number of iterations), then play the last iterate $x_t^{(K+1)}$ for $F_t$. To obtain the point $x_t^{(K+1)}$ which we play, we need to solve the linear program Eq. (1) and thus get $v_t^{(k)}$, where we have to know the gradient in advance. However, in the online setting, we can only observe the stochastic gradient $\tilde{\nabla} F_t$ *after* we play some point for $F_t$. So the key issue is to obtain the vector $v_t^{(k)}$ which at least approximately maximizes $\langle \cdot, \nabla F_t(x_t^{(k)}) \rangle$, *before* we play some point for $F_t$.

To do so, we use $K$ no-regret online linear maximization oracles $\{\mathcal{E}^{(k)}\}, k \in [K]$, and let $v_t^{(k)}$ be the output vector of $\mathcal{E}^{(k)}$ at round $t$. Once we update $x_t^{(k+1)}$ by $v_t^{(k)}$ for all $k \in [K]$, and play $x_t^{(K+1)}$ for $F_t$, we can observe $\tilde{\nabla} F_t(x_t^{(k)})$ and iteratively construct $d_t^{(k)} = (1 - \rho_k)d_t^{(k-1)} + \rho_k \tilde{\nabla} F_t(x_t^{(k)})$, an estimation of $\nabla F_t(x_t^{(k)})$ with a lower variance than $\tilde{\nabla} F_t(x_t^{(k)})$ [37, 38] for all $k \in [K]$. Then we set $\langle \cdot, d_t^{(k)} \rangle$ as the objective function for oracle $\mathcal{E}^{(k)}$ at round $t$. Thanks to the no-regret property of $\mathcal{E}^{(k)}$,

$v_t^{(k)}$, which is obtained before we play some point for $F_t$ and observe the gradient, approximately maximizes $\langle \cdot, d_t^{(k)} \rangle$, thus also approximately maximizes $\langle \cdot, \nabla F_t(x_t^{(k)}) \rangle$.

This approach was first proposed in [18, 17], where stochastic gradients at $K = T^{3/2}$ points (*i.e.*, $\{x_t^{(k)}\}, k \in [K]$) are required for each function $F_t$. To carry this general idea into the one-shot setting where we can only access one gradient per function, we need the following blocking procedure.

We divide the upcoming objective functions $F_1, \dots, F_T$ into $Q$ equisized blocks of size $K$ (so $T = QK$). For the $q$-th block, we first set $x_q^{(1)} = 0$, update $x_q^{(k+1)} = x_q^{(k)} + \eta_k v_q^{(k)}$, and play the same point $x_q = x_q^{(K+1)}$ for all the functions $F_{(q-1)K+1}, \dots, F_{qK}$. The reason why we play the same point $x_q$ will be explained later. We also define the average function in the $q$-th block as $\bar{F}_q \triangleq \frac{1}{K} \sum_{k=1}^{K} F_{(q-1)K+k}$. In order to reduce the required number of gradients per function, the key idea is to view the average functions $\bar{F}_1, \dots, \bar{F}_Q$ as *virtual* objective functions.

Precisely, in the $q$-th block, let $(t_{q,1}, \dots, t_{q,K})$ be a random permutation of the indices $\{(q-1)K + 1, \dots, qK\}$. *After* we update all the $x_q^{(k)}$, for each $F_t$, we play $x_q$ and find the corresponding $k'$ such that $t = t_{q,k'}$, then observe $\tilde{\nabla} F_t$ (*i.e.*, $\tilde{\nabla} F_{t_{q,k'}}$) at $x_q^{(k')}$. Thus we can obtain $\tilde{\nabla} F_{t_{q,k}}(x_q^{(k)})$ for all $k \in [K]$. Since $t_{q,k}$ is a random variable such that $\mathbb{E}[F_{t_{q,k}}] = \bar{F}_q$, $\tilde{\nabla} F_{t_{q,k}}(x_q^{(k)})$ is also an estimation of $\nabla \bar{F}_q(x_q^{(k)})$, which holds for all $k \in [K]$. As a result, with only one gradient evaluation per function $F_{t_{q,k}}$, we can obtain stochastic gradients of the virtual objective function $\bar{F}_q$ at $K$ points. In this way, the required number of per-function gradient evaluations is reduced from $K$ to 1 successfully.

Note that since we play $y_t = x_q$ for each $F_t$ in the $q$-th block, the regret w.r.t. the original objective functions and that w.r.t. the average functions satisfy that

$$(1 - 1/e) \max_{x \in \mathcal{K}} \sum_{t=1}^{T} F_t(x) - \sum_{t=1}^{T} F_t(y_t) = K \left[ (1 - 1/e) \max_{x \in \mathcal{K}} \sum_{q=1}^{Q} \bar{F}_q(x) - \sum_{t=1}^{Q} \bar{F}_q(x_q) \right],$$

which makes it possible to view the functions $\bar{F}_q$ as *virtual* objective functions in the regret analysis. Moreover, we iteratively construct $d_q^{(k)} = (1 - \rho_k) d_q^{(k-1)} + \rho_k \tilde{\nabla} F_{t_{q,k}}(x_q^{(k)})$ as an estimation of $\nabla F_{t_{q,k}}(x_q^{(k)})$, thus also an estimation of $\nabla \bar{F}_q(x_q^{(k)})$. So $v_q^{(k)}$, the output of $\mathcal{E}^{(k)}$, approximately maximizes $\langle \cdot, \nabla \bar{F}_q(x_q^{(k)}) \rangle$. Inspired by the offline FW method, playing $x_q = x_q^{(K+1)}$, the last iterate in the FW procedure, may obtain high rewards for $\bar{F}_q$. As a result, we play the same point $x_q$ in the $q$-th block.

We also note that once $t_{q,1}, \dots, t_{q,k}$ are revealed, conditioned on the knowledge, the expectation of $F_{t_{q,k+1}}$ is no longer the average function $\bar{F}_q$ but the residual average function $\bar{F}_{q,k}(x) = \frac{1}{K-k} \sum_{i=k+1}^{K} F_{t_{q,i}}(x)$. As more indices $t_{q,k}$ are revealed, $\bar{F}_{q,k}$ becomes increasingly different from $\bar{F}_q$, which makes the observed gradient $\tilde{\nabla} F_{t_{q,k+1}}(x_q^{(k+1)})$ not a good estimation of $\nabla \bar{F}_q(x_q^{(k+1)})$ any more. As a result, although we use the averaging technique (the update of $d_q^{(k)}$) as in [37, 38] for variance reduction, a *completely different* gradient error analysis is required. In Lemma 6 (Appendix B), we establish that the squared error of $d_q^{(k)}$ exhibits an inverted bell-shaped tendency; *i.e.*, the squared error is large at the initial and final stages and is small at the intermediate stage.

We present our proposed `Mono-Frank-Wolfe` algorithm in Algorithm 1.

We will show that `Mono-Frank-Wolfe` achieves a $(1 - 1/e)$-regret bound of $O(T^{4/5})$. In order to prove this result, we first make the following assumptions on the constraint set $\mathcal{K}$, the objective functions $F_t$, the stochastic gradient $\tilde{\nabla} F_t$, and the online linear maximization oracles.

**Assumption 1.** *The constraint set $\mathcal{K}$ is a convex and compact set that contains 0.*

**Assumption 2.** *Every objective function $F_t$ is monotone, continuous DR-Submodular, $L_1$-Lipschitz, and $L_2$-smooth.*

**Assumption 3.** *The stochastic gradient $\tilde{\nabla} F_t(x)$ is unbiased,* i.e., $\mathbb{E}[\tilde{\nabla} F_t(x)] = \nabla F_t(x)$. *Additionally, it has a uniformly bounded norm $\|\tilde{\nabla} F_t(x)\| \le M_0$ and a uniformly bounded variance $\mathbb{E}[\|\nabla F_t(x) - \tilde{\nabla} F_t(x)\|^2] \le \sigma_0^2$ for every $x \in \mathcal{K}$ and objective function $F_t$.*

---

**Algorithm 1** `Mono-Frank-Wolfe`

---

**Input:** constraint set $\mathcal{K}$, horizon $T$, block size $K$, online linear maximization oracles on $\mathcal{K}$:
$\quad$ $\mathcal{E}^{(1)}, \cdots, \mathcal{E}^{(K)}$, step sizes $\rho_k \in (0,1), \eta_k \in (0,1)$, number of blocks $Q = T/K$

**Output:** $y_1, y_2, \ldots$
 1: **for** $q = 1, 2, \ldots, Q$ **do**
 2: $\quad$ $d_q^{(0)} \leftarrow 0, x_q^{(1)} \leftarrow 0$
 3: $\quad$ For $k = 1, 2, \ldots, K$, let $v_q^{(k)} \in \mathcal{K}$ be the output of $\mathcal{E}^{(k)}$ in round $q$, $x_q^{(k+1)} \leftarrow x_q^{(k)} + \eta_k v_q^{(k)}$.
 $\quad\quad$ Set $x_q \leftarrow x_q^{(K+1)}$
 4: $\quad$ Let $(t_{q,1}, \ldots, t_{q,K})$ be a random permutation of $\{(q-1)K+1, \ldots, qK\}$
 5: $\quad$ For $t = (q-1)K+1, \ldots, qK$, play $y_t = x_q$ and obtain the reward $F_t(y_t)$; find the
 $\quad\quad$ corresponding $k' \in [K]$ such that $t = t_{q,k'}$, observe $\tilde{\nabla} F_t(x_q^{(k')})$, i.e., $\tilde{\nabla} F_{t_{q,k'}}(x_q^{(k')})$
 6: $\quad$ For $k = 1, 2, \ldots, K$, $d_q^{(k)} \leftarrow (1-\rho_k)d_q^{(k-1)} + \rho_k \tilde{\nabla} F_{t_{q,k}}(x_q^{(k)})$, compute $\langle v_q^{(k)}, d_q^{(k)} \rangle$ as
 $\quad\quad$ reward for $\mathcal{E}^{(k)}$, and feed back $d_q^{(k)}$ to $\mathcal{E}^{(k)}$
 7: **end for**

---

**Assumption 4.** *For the online linear maximization oracles, the regret at horizon $t$ (denoted by $\mathcal{R}_t^{\mathcal{E}^{(i)}}$) satisfies $\mathcal{R}_t^{\mathcal{E}^{(i)}} \leq C\sqrt{t}, \forall i \in [K]$, where $C > 0$ is a constant.*

Note that there exist online linear maximization oracles $\mathcal{E}^{(i)}$ with regret $\mathcal{R}_t^{\mathcal{E}^{(i)}} \leq C\sqrt{t}, \forall i \in [K]$ for any horizon $t$ (for example, the online gradient descent [50]). Therefore, Assumption 4 is fulfilled.

**Theorem 1** (Proof in Appendix B). *Under Assumptions 1 to 4, if we set $K = T^{3/5}, \eta_k = \frac{1}{K}, \rho_k = \frac{2}{(k+3)^{2/3}}$ when $1 \leq k \leq K/2+1$, and $\rho_k = \frac{1.5}{(K-k+2)^{2/3}}$ when $K/2+2 \leq k \leq K$, where we assume that $K$ is even for simplicity, then $y_t \in \mathcal{K}, \forall t$, and the expected $(1-1/e)$-regret of Algorithm 1 is at most*

$$\mathbb{E}[\mathcal{R}_T] \leq (N + C + D^2)T^{4/5} + \frac{L_2 D^2}{2} T^{2/5},$$

*where $N = \max\{5^{2/3}(L_1 + M_0)^2, 4(L_1^2 + \sigma_0^2) + 32G, 2.25(L_1^2 + \sigma_0^2) + 7G/3\}, G = (L_2 R + 2L_1)^2$.*

## 4 Bandit Continuous DR-Submodular Maximization

In this section, we present the first bandit algorithm for continuous DR-submodular maximization, `Bandit-Frank-Wolfe`, which attains a $(1-1/e)$-regret bound of $O(T^{8/9})$. We begin by explaining the one-point gradient estimator [23], which is crucial to the proposed bandit algorithm. The proposed algorithm and main results are illustrated in Section 4.2.

### 4.1 One-Point Gradient Estimator

Given a function $F$, we define its $\delta$-smoothed version $\hat{F}_\delta(x) \triangleq \mathbb{E}_{v \sim B^d}[F(x + \delta v)]$, where $v \sim B^d$ denotes that $v$ is drawn uniformly at random from the unit ball $B^d$. Thus the function $F$ is averaged over a ball of radius $\delta$. It can be easily verified that if $F$ is monotone, continuous DR-submodular, $L_1$-Lipschitz, and $L_2$-smooth, then so is $\hat{F}_\delta$, and for all $x$ we have $|\hat{F}_\delta(x) - F(x)| \leq L_1 \delta$ (Lemma 7 in Appendix C). So the $\delta$-smoothed version $\hat{F}_\delta$ is indeed an approximation of $F$. A maximizer of $\hat{F}_\delta$ also maximizes $F$ approximately.

More importantly, the gradient of the smoothed function $\tilde{F}_\delta$ admits a one-point unbiased estimator [23, 32]: $\nabla \hat{F}_\delta(x) = \mathbb{E}_{u \sim S^{d-1}} \left[\frac{d}{\delta} F(x + \delta u)u\right]$, where $u \sim S^{d-1}$ denotes that $u$ is drawn uniformly at random from the unit sphere $S^{d-1}$. Thus the player can estimate the gradient of the smoothed function at point $x$ by playing the random point $x + \delta u$ for the original function $F$. So usually, we can extend a one-shot online algorithm to the bandit setting by replacing the observed stochastic gradients with the one-point gradient estimations.

In our setting, however, we cannot use the one-point gradient estimator directly. When the point $x$ is close to the boundary of the constraint set $\mathcal{K}$, the point $x + \delta u$ may fall outside of $\mathcal{K}$. To address this

issue, we introduce the notion of $\delta$-*interior*. A set is said to be a $\delta$-interior of $\mathcal{K}$ if it is a *subset* of

$$\text{int}_\delta(\mathcal{K}) = \{x \in \mathcal{K} \mid \inf_{s \in \partial\mathcal{K}} d(x,s) \geq \delta\},$$

where $d(\cdot,\cdot)$ denotes the Euclidean distance.

In other words, $\mathcal{K}'$ is a $\delta$-interior of $\mathcal{K}$ if it holds for every $x \in \mathcal{K}'$ that $B(x,\delta) \subseteq \mathcal{K}$ (Fig. 1a in Appendix D). We note that there can be infinitely many $\delta$-interiors of $\mathcal{K}$. In the sequel, $\mathcal{K}'$ will denote the $\delta$-interior that we consider. We also define the discrepancy between $\mathcal{K}$ and $\mathcal{K}'$ by

$$d(\mathcal{K},\mathcal{K}') = \sup_{x \in \mathcal{K}} d(x,\mathcal{K}'),$$

which is the supremum of the distances between points in $\mathcal{K}$ and the set $\mathcal{K}'$. The distance $d(x,\mathcal{K}')$ is given by $\inf_{y \in \mathcal{K}'} d(x,y)$.

By definition, every point $x \in \mathcal{K}'$ satisfies $x + \delta u \in \mathcal{K}$, which enables us to use the one-point gradient estimator on $\mathcal{K}'$. Moreover, if every $F_t$ is Lipschitz and $d(\mathcal{K},\mathcal{K}')$ is small, we can approximate the optimal total reward on $\mathcal{K}$ ($\max_{x \in \mathcal{K}} \sum_{t=1}^T F_t(x)$) by that on $\mathcal{K}'$ ($\max_{x \in \mathcal{K}'} \sum_{t=1}^T F_t(x)$), and thereby obtain the regret bound subject to the original constraint set $\mathcal{K}$, by running bandit algorithms on $\mathcal{K}'$.

We also note that if the constraint set $\mathcal{K}$ satisfies Assumption 1 and is down-closed (*e.g.*, a matroid polytope), for sufficiently small $\delta$, we can *construct* $\mathcal{K}'$, a down-closed $\delta$-interior of $\mathcal{K}$, with $d(\mathcal{K},\mathcal{K}')$ sufficiently small (actually it is a linear function of $\delta$). Recall that a set $\mathcal{P}$ is down-closed if it has a lower bound $\underline{u}$ such that (1) $\forall y \in \mathcal{P}, \underline{u} \leq y$; and (2) $\forall y \in \mathcal{P}, x \in \mathbb{R}^d, \underline{u} \leq x \leq y \implies x \in \mathcal{P}$ [9].

We first define $B_{\geq 0}^d = B^d \cap \mathbb{R}_{\geq 0}^d$ and make the following assumption[2]:

**Assumption 5.** *There exists a positive number $r$ such that $r B_{\geq 0}^d \subseteq \mathcal{K}$.*

To construct $\mathcal{K}'$, for sufficiently small $\delta$ such that $\delta < \frac{r}{\sqrt{d}+1}$, we first set $\alpha = \frac{(\sqrt{d}+1)\delta}{r} < 1$, and shrink $\mathcal{K}$ by a factor of $(1-\alpha)$ to obtain $\mathcal{K}_\alpha = (1-\alpha)\mathcal{K}$. Then we translate the shrunk set $\mathcal{K}_\alpha$ by $\delta\mathbf{1}$ (Fig. 1b in Appendix D). In other words, the set that we finally obtain is

$$\mathcal{K}' = \mathcal{K}_\alpha + \delta\mathbf{1} = (1-\alpha)\mathcal{K} + \delta\mathbf{1}.$$

In Lemma 1, we establish that $\mathcal{K}'$ is indeed a $\delta$-interior of $\mathcal{K}$ and deduce a linear bound for $d(\mathcal{K},\mathcal{K}')$.

**Lemma 1** (Proof in Appendix D). *We assume Assumptions 1 and 5 and also assume that $\mathcal{K}$ is down-closed and that $\delta$ is sufficiently small such that $\alpha = \frac{(\sqrt{d}+1)\delta}{r} < 1$. The set $\mathcal{K}' = (1-\alpha)\mathcal{K} + \delta\mathbf{1}$ is convex and compact. Moreover, $\mathcal{K}'$ is a down-closed $\delta$-interior of $\mathcal{K}$ and satisfies $d(\mathcal{K},\mathcal{K}') \leq [\sqrt{d}(\frac{R}{r}+1) + \frac{R}{r}]\delta$.*

### 4.2 No-$(1-1/e)$-Regret Biphasic Bandit Algorithm

Our proposed bandit algorithm is based on the online algorithm `Mono-Frank-Wolfe` in Section 3. Precisely, we want to replace the stochastic gradients in Algorithm 1 with the one-point gradient estimators, and run the modified algorithm on $\mathcal{K}'$, a proper $\delta$-interior of the constraint set $\mathcal{K}$. Note that the one-point estimator requires that the point at which we estimate the gradient (*i.e.*, $x$) must be identical to the point that we play (*i.e.*, $x + \delta u$), if we ignore the random $\delta u$. In Algorithm 1, however, we play point $x_q$ but obtain estimated gradient at other points $x_q^{(k')}$ (Line 5). This suggests that Algorithm 1 cannot be extended to the bandit setting via the one-point gradient estimator directly.

To circumvent this limitation, we propose a biphasic approach that categorizes the plays into the exploration and exploitation phases. To motivate this biphasic method, recall that in Algorithm 1, we need to play $x_q$ to gain high rewards (exploitation), whilst we observe $\tilde{\nabla} F_t(x_q^{(k')})$ to obtain gradient information (exploration). So in our biphasic approach, we expend a large portion of plays on exploitation (play $x_q$, so we can still get high rewards) and a small portion of plays on exploring the gradient (play $x_q^{(k')}$ to get one-point gradient estimators, so we can still obtain sufficient information).

To be precise, we divide the $T$ objective functions into $Q$ equisized blocks of size $L$, where $L = T/Q$. Each block is subdivided into two phases. As shown in Algorithm 2, we randomly choose $K \ll L$ functions for exploration (Line 6) and use the remaining $(L - K)$ functions for exploitation (Line 7).

We describe our algorithm formally in Algorithm 2. We also note that for a general constraint set $\mathcal{K}$ with a proper $\delta$-interior $\mathcal{K}'$ such that $d(\mathcal{K}, \mathcal{K}') \leq c_1 \delta^\gamma$, *Theorem 4* (Appendix E.1) shows a $(1 - 1/e)$-regret bound of $O(T^{\frac{3+5\min\{1,\gamma\}}{3+6\min\{1,\gamma\}}})$. Moreover, with Lemma 1, this result can be extended to down-closed constraint sets $\mathcal{K}$, as shown in Theorem 2.

---

**Algorithm 2** `Bandit-Frank-Wolfe`

---

**Input:** smoothing radius $\delta$, $\delta$-interior $\mathcal{K}'$ with lower bound $\underline{u}$, horizon $T$, block size $L$, the number of exploration steps per block $K$, online linear maximization oracles on $\mathcal{K}'$: $\mathcal{E}^{(1)}, \cdots, \mathcal{E}^{(K)}$, step sizes $\rho_k \in (0,1), \eta_k \in (0,1)$, the number of blocks $Q = T/L$

**Output:** $y_1, y_2, \ldots$

1: **for** $q = 1, 2, \ldots, Q$ **do**
2:      $d_q^{(0)} \leftarrow 0, x_q^{(1)} \leftarrow \underline{u}$
3:      For $k = 1, 2, \ldots, K$, let $v_q^{(k)} \in \mathcal{K}'$ be the output of $\mathcal{E}^{(k)}$ in round $q$, $x_q^{(k+1)} \leftarrow x_q^{(k)} + \eta_k(v_q^{(k)} - \underline{u})$. Set $x_q \leftarrow x_q^{(K+1)}$
4:      Let $(t_{q,1}, \ldots, t_{q,L})$ be a random permutation of $\{(q-1)L + 1, \cdots, qL\}$
5:      **for** $t = (q-1)L + 1, \cdots, qL$ **do**
6:          If $t \in \{t_{q,1}, \cdots, t_{q,K}\}$, find the corresponding $k' \in [K]$ such that $t = t_{q,k'}$, play $y_t = y_{t_{q,k'}} = x_q^{(k')} + \delta u_{q,k'}$ for $F_t$ (*i.e.*, $F_{t_{q,k'}}$), where $u_{q,k'} \sim S^{d-1}$     ▷ *Exploration*
7:          If $t \in \{(q-1)L + 1, \cdots, qL\} \setminus \{t_{q,1}, \cdots, t_{q,K}\}$, play $y_t = x_q$ for $F_t$     ▷ *Exploitation*
8:      **end for**
9:      For $k = 1, 2, \ldots, K$, $g_{q,k} \leftarrow \frac{d}{\delta} F_{t_{q,k}}(y_{t_{q,k}}) u_{q,k}$, $d_q^{(k)} \leftarrow (1 - \rho_k) d_q^{(k-1)} + \rho_k g_{q,k}$, compute $\langle v_q^{(k)}, d_q^{(k)} \rangle$ as reward for $\mathcal{E}^{(k)}$, and feed back $d_q^{(k)}$ to $\mathcal{E}^{(k)}$
10: **end for**

---

**Assumption 6.** *Every objective function $F_t$ satisfies that $\sup_{x \in \mathcal{K}} |F_t(x)| \leq M_1$.*

**Theorem 2** (Proof in Appendix E.2)**.** *We assume Assumptions 1, 2 and 4 to 6, and also assume that $\mathcal{K}$ is down-closed. If we generate $\mathcal{K}'$ as in Lemma 1, and set $\delta = \frac{r}{\sqrt{d}+2} T^{-\frac{1}{9}}, L = T^{\frac{7}{9}}, K = T^{\frac{2}{3}}, \eta_k = \frac{1}{K}, \rho_k = \frac{2}{(k+2)^{2/3}}$, then $y_t \in \mathcal{K}, \forall t$, and the expected $(1 - 1/e)$-regret of Algorithm 2 is at most*

$$\mathbb{E}[\mathcal{R}_T] \leq NT^{\frac{8}{9}} + \frac{3r[2L_1^2 + (3L_2 R + 2L_1)^2]}{4^{1/3}(\sqrt{d}+2)} T^{\frac{2}{3}} + \frac{L_2 D^2}{2} T^{\frac{1}{3}},$$

*where $N = \frac{(1-1/e)r}{\sqrt{d}+2}[\sqrt{d}(\frac{R}{r}+1) + \frac{R}{r}]L_1 + \frac{(2-1/e)r}{\sqrt{d}+2}L_1 + 2M_1 + \frac{3 \cdot 4^{1/6}(\sqrt{d}+2)d^2 M_1^2}{r} + \frac{3(\sqrt{d}+2)D^2}{4r} + C.$*

## 5 Bandit Submodular Set Maximization

In this section we aim to solve the problem of bandit submodular set maximization by lifting it to the continuous domain. Let objective functions $f_1, \cdots, f_T : 2^\Omega \to \mathbb{R}_{\geq 0}$ be a sequence of monotone submodular set functions defined on a common ground set $\Omega = \{1, \ldots, d\}$. We also let $\mathcal{I}$ denote the matroid constraint, and $\mathcal{K}$ be the matroid polytope of $\mathcal{I}$, *i.e.*, $\mathcal{K} = \text{conv}\{\mathbf{1}_I : I \in \mathcal{I}\} \subseteq [0,1]^d$ [16], where conv denotes the convex hull.

### 5.1 An Impossibility Result

A natural idea is that at each round $t$, we apply `Bandit-Frank-Wolfe`, the continuous algorithm in Section 4.2, on $F_t$ subject to $\mathcal{K}$, where $F_t$ is the multilinear extension of the discrete objective function $f_t$. Then we get a fractional solution $y_t \in \mathcal{K}$, round it to a set $Y_t \in \mathcal{I}$, and play $Y_t$ for $f_t$.

For the exploitation phase, we will use a lossless rounding scheme such that $f_t(Y_t) \geq F_t(y_t)$, so we will not get lower rewards after the rounding. Instances of such a lossless rounding scheme include pipage rounding [4, 16] and the contention resolution scheme [46].

In the exploration phase, we need to use the reward $f_t(Y_t)$ to obtain an unbiased gradient estimator of the smoothed version of $F_t$. As the one-point estimator $\frac{d}{\delta} F(x + \delta u) u$ in Algorithm 2 is unbiased, we

require the (random) rounding scheme $\text{round}_\mathcal{I} : [0,1]^d \to \mathcal{I}$ to satisfy the following unbiasedness condition

$$\mathbb{E}[f(\text{round}_\mathcal{I}(x))] = F(x), \quad \forall x \in [0,1]^d \tag{2}$$

for any submodular set function $f$ on the ground set $\Omega$ and its multilinear extension $F$.

Since we have no *a priori* knowledge of the objective function $f_t$ before playing a subset for it, such a rounding scheme $\text{round}_\mathcal{I}$ should *not* depend on the function choice $f$. In other words, we need to find an *independent* $\text{round}_\mathcal{I}$ such that Eq. (2) holds for any submodular function $f$ defined on $\Omega$.

We first review the random rounding scheme $\text{RandRound} : [0,1]^d \to \mathcal{I}$

$$\begin{cases} i \in \text{RandRound}(x) & \text{with probability } x_i \, ; \\ i \notin \text{RandRound}(x) & \text{with probability } 1 - x_i \, . \end{cases} \tag{3}$$

In other words, each element $i \in \Omega$ is included with an independent probability $x_i$, where $x_i$ is the $i$-th coordinate of $x$. $\text{RandRound}$ satisfies the unbiasedness requirement Eq. (2). However, its range is $2^\Omega$ in general, so the rounded set may fall outside of $\mathcal{I}$. In fact, as shown in Lemma 2, there exists a matroid $\mathcal{I}$ for which we cannot find a proper unbiased rounding scheme whose range is contained in $\mathcal{I}$.

**Lemma 2** (Proof in Appendix F). *There exists a matroid $\mathcal{I}$ for which there is no rounding scheme* $\text{round} : [0,1]^d \to \mathcal{I}$ *whose construction does not depend on the function $f$ and which satisfies Eq. (2) for any submodular set function $f$.*

## 5.2 Responsive Bandit Algorithm

The impossibility result Lemma 2 shows that the one-point estimator may be incapable of solving the general BSM problem. As a result, we study a slightly relaxed setting termed the responsive bandit submodular maximization problem (RBSM). Let $X_t$ be the subset that we play at the $t$-th round. The only difference between the responsive bandit setting and the vanilla bandit setting is that in the responsive setting, if $X_t \notin \mathcal{I}$, we can still observe the function value $f_t(X_t)$ as feedback, while the received reward at round $t$ is 0 (since the subset that we play violates the constraint $\mathcal{I}$). In other words, the environment is always responsive to the player's decisions, no matter whether $X_t$ is in $\mathcal{I}$ or not.

We note that the RBSM problem has broad applications in both theory and practice. In theory, RBSM can be regarded as a relaxation of BSM, which helps us to better understand the nature of BSM. In practice, the responsive model (not only for submodular maximization or bandit) has potentially many applications when a decision cannot be committed, while we can still get the potential outcome of the decision as feedback. For example, suppose that we have a replenishable inventory of items where customers arrive (in an online fashion) with a utility function unknown to us. We need to allocate a collection of items to each customer, and the goal is to maximize the total utility (reward) of all the customers. We may use a partition matroid to model diversity (in terms of category, time, *etc*). In the RBSM model, we cannot allocate the collection of items which violates the constraint to the customer, but we can use it as a questionnaire, and the customer will tell us the potential utility if she received those items. The feedback will help us to make better decisions in the future. Similar examples include portfolio selection when the investment choice is too risky, *i.e.*, violates the recommended constraint set, we may stop trading and thus get no reward on that trading period, but at the same time observe the potential reward if we invested in that way.

Now, we turn to propose our algorithm. As discussed in Section 5.1, we want to solve the problem of bandit submodular set maximization by applying Algorithm 2 on the multilinear extensions $F_t$ with different rounding schemes. Precisely, in the *responsive* setting, we use the $\text{RandRound}$ Eq. (3) in the exploration phase to guarantee that we can always obtain unbiased gradient estimators, and use a lossless rounding scheme $\text{LosslessRound}$ in the exploitation phase to receive high rewards. We present `Responsive-Frank-Wolfe` in Algorithm 3, and show that it achieves a $(1 - 1/e)$-regret bound of $O(T^{8/9})$.

**Assumption 7.** *Every objective function $f_t$ is monotone submodular with $\sup_{X \subseteq \Omega} |f_t(X)| \le M_1$.*

**Theorem 3** (Proof in Appendix G). *Under Assumptions 4, 5 and 7, if we generate $\mathcal{K}'$ as in Lemma 1, and set $\delta = \frac{r}{\sqrt{d}+2} T^{-\frac{1}{9}}, L = T^{\frac{7}{9}}, K = T^{\frac{2}{3}}, \eta_k = \frac{1}{K}, \rho_k = \frac{2}{(k+2)^{2/3}}$, then in the responsive setting,*

---

**Algorithm 3** `Responsive-Frank-Wolfe`

---

**Input:** matroid constraint $\mathcal{I}$, matroid polytope $\mathcal{K}$, smoothing radius $\delta$, $\delta$-interior $\mathcal{K}'$ with lower bound $\underline{u}$, horizon $T$, block size $L$, the number of exploration steps per block $K$, online linear maximization oracles on $\mathcal{K}'$: $\mathcal{E}^{(1)}, \cdots, \mathcal{E}^{(K)}$, steps sizes $\rho_k \in (0,1), \eta_k \in (0,1)$, the number of blocks $Q = T/L$

**Output:** $Y_1, Y_2, \ldots$

 1: **for** $q = 1, 2, \ldots, Q$ **do**
 2: $\quad$ $d_q^{(0)} \leftarrow 0, x_q^{(1)} \leftarrow \underline{u}$
 3: $\quad$ For $k = 1, 2, \ldots, K$, let $v_q^{(k)} \in \mathcal{K}'$ be the output of $\mathcal{E}^{(k)}$ in round $q$, $x_q^{(k+1)} \leftarrow x_q^{(k)} + \eta_k(v_q^{(k)} - \underline{u})$. Set $x_q \leftarrow x_q^{(K+1)}$
 4: $\quad$ Let $(t_{q,1}, \ldots, t_{q,L})$ be a random permutation of $\{(q-1)L + 1, \cdots, qL\}$
 5: $\quad$ **for** $t = (q-1)L + 1, \cdots, qL$ **do**
 6: $\quad\quad$ If $t \in \{t_{q,1}, \cdots, t_{q,K}\}$, find the corresponding $k' \in [K]$ such that $t = t_{q,k'}$, play $Y_t = Y_{t_{q,k'}} = \text{RandRound}(y_{t_{q,k'}})$ for $f_t$ (*i.e.*, $f_{t_{q,k'}}$), where $y_{t_{q,k'}} = x_q^{(k')} + \delta u_{q,k'}, u_{q,k'} \sim S^{d-1}$. If $Y_t \in \mathcal{I}$, get reward $f_t(Y_t)$; otherwise, get reward 0. $\qquad\qquad\qquad$ ▷ *Exploration*
 7: $\quad\quad$ If $t \in \{(q-1)L + 1, \cdots, qL\} \setminus \{t_{q,1}, \cdots, t_{q,K}\}$, play $Y_t = \text{LosslessRound}(y_t)$ for $f_t$, where $y_t = x_q$ $\qquad\qquad\qquad\qquad\qquad\qquad\qquad\qquad\qquad\qquad\qquad$ ▷ *Exploitation*
 8: $\quad$ **end for**
 9: $\quad$ For $k = 1, 2, \ldots, K$, $g_{q,k} \leftarrow \frac{d}{\delta} f_{t_{q,k}}(Y_{t_{q,k}}) u_{q,k}$, $d_q^{(k)} \leftarrow (1-\rho_k)d_q^{(k-1)} + \rho_k g_{q,k}$, compute $\langle v_q^{(k)}, d_q^{(k)} \rangle$ as reward for $\mathcal{E}^{(k)}$, and feed back $d_q^{(k)}$ to $\mathcal{E}^{(k)}$
10: **end for**

---

*the expected $(1 - 1/e)$-regret of Algorithm 3 is at most*

$$\mathbb{E}[\mathcal{R}_T] \leq NT^{\frac{8}{9}} + \frac{3r[2L_1^2 + (3\sqrt{d}L_2 + 2L_1)^2]}{4^{1/3}(\sqrt{d}+2)} T^{\frac{2}{3}} + \frac{L_2 d}{2} T^{\frac{1}{3}},$$

*where* $N = \frac{(1-1/e)r}{\sqrt{d}+2}[\frac{d}{r} + \sqrt{d}(1+\frac{1}{r})]L_1 + \frac{(2-1/e)r}{\sqrt{d}+2} L_1 + 3M_1 + \frac{3 \cdot 4^{2/3}(\sqrt{d}+2)d^2 M_1^2}{r} + \frac{3(\sqrt{d}+2)d}{4r} + C$, $L_1 = 2M_1\sqrt{d}, L_2 = 4M_1\sqrt{d(d-1)}$.

## 6 Conclusion

In this paper, by proposing a series of novel methods including the *blocking procedure* and the *permutation methods*, we developed `Mono-Frank-Wolfe` for the OCSM problem, which requires only one stochastic gradient evaluation per function and still achieves a $(1-1/e)$-regret bound of $O(T^{4/5})$. We then introduced the *biphasic method* and the notion of $\delta$-*interior*, to extend `Mono-Frank-Wolfe` to `Bandit-Frank-Wolfe` for the BCSM problem. Finally, we introduced the responsive model and the corresponding `Responsive-Frank-Wolfe` Algorithm for the RBSM problem. We proved that both `Bandit-Frank-Wolfe` and `Responsive-Frank-Wolfe` attain a $(1 - 1/e)$-regret bound of $O(T^{8/9})$.

**Acknowledgments**

This work is partially supported by the Google PhD Fellowship, NSF (IIS-1845032), ONR (N00014-19-1-2406) and AFOSR (FA9550-18-1-0160). We would like to thank Marko Mitrovic for his valuable comments and Zheng Wei for help preparing some of the illustrations.

## Footnotes

[1]We formally define the multilinear extension of a submodular set function in Section 2.

[2]This assumption is an analogue of the assumption $rB^d \subseteq \mathcal{K} \subseteq RB^d$ in [23].

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
