[Supplementary Material]

## A  Further Related Work

The framework of online convex optimization (OCO) dates back to [50], where a regret bound of $O(\sqrt{T})$ was attained. The regret bound was improved to $\log(T)$ for strongly convex losses in [31]. The RFTL algorithm was proposed independently in [41, 42]. The projection-free algorithm Online Conditional Gradient was proposed in [28, 32]. The model of Bandit Convex Optimization (BCO) was introduced in [23], and followed by plenty of works [21, 3, 13, 11]. Various regret bounds were achieved by adding extra assumptions (*e.g.*, strong convexity) in [35, 2, 40, 29, 14, 22, 30, 15]. The first computationally efficient projection-free BCO algorithm was proposed in [20]. For strongly convex and smooth losses, a lower bound of $\Omega(\sqrt{T})$ for regret was proved in [43]. Bandit linear optimization was studied in [1, 5, 12]. Interested readers are referred to [13] for a survey on BCO.

Bach [6] derived connections between continuous submodularity and convexity. Bian et al. [9] studied the offline continuous DR-submodular maximization and proposed a variant of the Frank-Wolfe algorithm to achieve the tight $(1 - 1/e)$ approximation ratio. In the online setting, maximization of submodular set functions was studied in [44, 27]. Adaptive submodular bandit maximization was analyzed in [25]. The linear submodular bandit problems were studied in [49, 48].

## B  Proof of Theorem 1

*Proof.* Since $y_t = x_q = x_q^{(K+1)}$, which is a convex combination of $v_q^{(1)}, v_q^{(2)}, \cdots, v_q^{(K)}$, and $v_q^{(k)} \in \mathcal{K}, \forall k \in [K]$, we have $y_t \in \mathcal{K}$. Then we proceed to prove the theorem.

The key idea of Algorithm 1 is to use the average function of a bunch of functions in certain group (*e.g.*, the block) to represent the functions. Note the regret is calculated by the sum of all the reward functions, and the sum of average functions is exactly the sum of all the functions divided by the block size, so we can use the average function to analyze the regret.

Let

$$\bar{F}_{q,k}(x) = \frac{\sum_{i=k+1}^{K} F_{t_{q,i}}(x)}{K - k}, k \in \{0, 1, \cdots, K - 1\}$$

denotes the average function of the remaining $(K - k)$ functions after round $k$ in the $q$-th block. Recall that $(t_{q,1}, \ldots, t_{q,K})$ is a random permutation of $((q-1)K, qK] \cap \mathbb{Z}$, thus $\bar{F}_{q,k}(x)$ is a random function. Also, by definition, we have the expected regret

$$\mathbb{E}[\sum_{t=1}^{T}(1 - 1/e)F_t(x^*) - F_t(x_q)] = \mathbb{E}[\sum_{q=1}^{Q} K[(1 - 1/e)\bar{F}_{q,0}(x^*) - \bar{F}_{q,0}(x_q)]], \qquad (4)$$

where $x^* = \arg\max_{x \in \mathcal{K}} \sum_{t=1}^{T} F_t(x)$. We also note that on the left hand side of Eq. (4), $q$ is actually a function of $t$. Specifically , $q$ is the index of the block which contains $F_t$.

**Lemma 3** (Eq.(9) in [17])**.** *If $F_t$ is monotone continuous DR-submodular and $L_2$-smooth, $x_t^{(k+1)} = x_t^{(k)} + 1/K \cdot v_t^{(k)}$ for $k \in [K]$, then*

$$F_t(x^*) - F_t(x_t^{(k+1)}) \leq (1 - 1/K)[F_t(x^*) - F_t(x_t^{(k)})]$$

$$- \frac{1}{K}[-\frac{1}{2\beta^{(k)}}\|\nabla F_t(x_t^{(k)}) - d_t^{(k)}\|^2 - \frac{\beta^{(k)}D^2}{2} + \langle d_t^{(k)}, v_t^{(k)} - x^*\rangle] + \frac{L_2 D^2}{2K^2},$$

*where $\{\beta^{(k)}\}$ is a sequence of positive parameters to be determined.*

**Lemma 4.** *If $F_t$ is monotone continuous DR-submodular and $L_2$-smooth for all $t$, $x_q^{(k+1)} = x_q^{(k)} + 1/K \cdot v_q^{(k)}$ for $k \in [K]$, and $x_q = x_q^{(K+1)}$, then we have*

$$\mathbb{E}[(1 - 1/e)\bar{F}_{q,0}(x^*) - \bar{F}_{q,0}(x_q)] \leq \mathbb{E}[\frac{1}{K}\sum_{k=1}^{K}[\frac{1}{2\beta^{(k)}}\Delta_q^{(k)} + \frac{\beta^{(k)}D^2}{2}]] + \frac{L_2 D^2}{2K}$$

$$+ 1/K \sum_{k=1}^{K}(1 - 1/K)^{K-k}\mathbb{E}[\langle d_q^{(k)}, x^* - v_q^{(k)}\rangle],$$

*where* $\Delta_q^{(k)} = \|\nabla \bar{F}_{q,k-1}(x_q^{(k)}) - d_q^{(k)}\|^2.$

*Proof of Lemma 4.* Since $F_t$ is monotone continuous DR-Submodular and $L_2$-smooth, then so is $\bar{F}_{q,k-1}$. By Lemma 3, we have

$$
\begin{aligned}
\mathbb{E}[\bar{F}_{q,0}(x^*) - \bar{F}_{q,0}(x_q^{(k+1)})] =& \mathbb{E}[\bar{F}_{q,k-1}(x^*) - \bar{F}_{q,k-1}(x_q^{(k+1)})] \\
\leq& \mathbb{E}[(1 - 1/K)[\bar{F}_{q,k-1}(x^*) - \bar{F}_{q,k-1}(x_q^{(k)})] + \frac{L_2 D^2}{2K^2} \\
& - \frac{1}{K}[-\frac{1}{2\beta^{(k)}}\|\nabla \bar{F}_{q,k-1}(x_q^{(k)}) - d_q^{(k)}\|^2 - \frac{\beta^{(k)} D^2}{2} + \langle d_q^{(k)}, v_q^{(k)} - x^* \rangle]].
\end{aligned}
\tag{5}
$$

Note that $\mathbb{E}\left[\bar{F}_{q,k-1}(x^*) - \bar{F}_{q,k-1}(x_q^{(k)})\right] = \mathbb{E}[\bar{F}_{q,k-2}(x^*) - \bar{F}_{q,k-2}(x_q^{(k)})]$, so we can apply Eq. (5) recursively for $k \in \{1, 2, \cdots, K\}$, and get

$$
\begin{aligned}
\mathbb{E}[\bar{F}_{q,0}(x^*) - \bar{F}_{q,0}(x_q)] \leq& \mathbb{E}[(1 - 1/K)^K[\bar{F}_{q,0}(x^*) - \bar{F}_{q,0}(x_q^{(1)})] + \frac{1}{K}\sum_{k=1}^{K}[\frac{1}{2\beta^{(k)}}\Delta_q^{(k)} + \frac{\beta^{(k)} D^2}{2}]] \\
& + \frac{L_2 D^2}{2K} + 1/K\sum_{k=1}^{K}(1 - 1/K)^{K-k}\mathbb{E}[\langle d_q^{(k)}, x^* - v_q^{(k)} \rangle],
\end{aligned}
$$

where $\Delta_q^{(k)} = \|\nabla \bar{F}_{q,k-1}(x_q^{(k)}) - d_q^{(k)}\|^2.$

Recall that $\bar{F}_{q,0}(x_q^{(1)}) = \bar{F}_{q,0}(0) \geq 0$ and $(1 - 1/K)^K \leq 1/e, \forall K \geq 1$, so we have

$$
\begin{aligned}
\mathbb{E}[(1 - 1/e)\bar{F}_{q,0}(x^*) - \bar{F}_{q,0}(x_q)] \leq& \mathbb{E}[\frac{1}{K}\sum_{k=1}^{K}[\frac{1}{2\beta^{(k)}}\Delta_q^{(k)} + \frac{\beta^{(k)} D^2}{2}]] + \frac{L_2 D^2}{2K} \\
& + 1/K\sum_{k=1}^{K}(1 - 1/K)^{K-k}\mathbb{E}[\langle d_q^{(k)}, x^* - v_q^{(k)} \rangle].
\end{aligned}
$$

$\square$

Combine Eq. (4) and Lemma 4, we have that the expected regret of Algorithm 1 satisfies:

$$
\begin{aligned}
\mathbb{E}[\mathcal{R}_T] =& \mathbb{E}[\sum_{t=1}^{T}(1 - 1/e)F_t(x^*) - F_t(x_q)] \\
=& \mathbb{E}[\sum_{q=1}^{Q} K[(1 - 1/e)\bar{F}_{q,0}(x^*) - \bar{F}_{q,0}(x_q)]] \\
\leq& \mathbb{E}[\sum_{q=1}^{Q}[\sum_{k=1}^{K}[\frac{1}{2\beta^{(k)}}\Delta_q^{(k)} + \frac{\beta^{(k)} D^2}{2}] + \frac{L_2 D^2}{2}]] + \sum_{q=1}^{Q}\sum_{k=1}^{K}(1 - 1/K)^{K-k}\mathbb{E}[\langle d_q^{(k)}, x^* - v_q^{(k)} \rangle] \\
=& \mathbb{E}[\sum_{q=1}^{Q}\sum_{k=1}^{K}\frac{1}{2\beta^{(k)}}\Delta_q^{(k)} + \frac{D^2}{2}Q\sum_{k=1}^{K}\beta^{(k)}] + \frac{L_2 D^2}{2}Q \\
& + \sum_{k=1}^{K}(1 - 1/K)^{K-k}\mathbb{E}[\sum_{q=1}^{Q}\langle d_q^{(k)}, x^* - v_q^{(k)} \rangle].
\end{aligned}
$$

Since $v_q^{(k)}$ is the output of the online linear maximization oracle $\mathcal{E}^{(k)}$ at round $q$, we have

$$
\sum_{q=1}^{Q}\langle d_q^{(k)}, x^* - v_q^{(k)} \rangle \leq \mathcal{R}_Q^{\mathcal{E}},
$$

and thus we have

$$\sum_{k=1}^{K}(1-1/K)^{K-k}\mathbb{E}[\sum_{q=1}^{Q}\langle d_q^{(k)}, x^* - v_q^{(k)}\rangle] \le \sum_{k=1}^{K} 1 \cdot \mathcal{R}_Q^{\mathcal{E}} = K\mathcal{R}_Q^{\mathcal{E}}.$$

Therefore,

$$\mathbb{E}[\mathcal{R}_T] \le \mathbb{E}[\sum_{q=1}^{Q}\sum_{k=1}^{K}\frac{1}{2\beta^{(k)}}\Delta_q^{(k)}] + \frac{D^2}{2}Q\sum_{k=1}^{K}\beta^{(k)} + K\mathcal{R}_Q^{\mathcal{E}} + \frac{L_2 D^2}{2}Q. \tag{6}$$

Note $\mathcal{R}_Q^{\mathcal{E}}$ is the regret of oracle $\mathcal{E}$ at horizon $Q$, which is of order $O(\sqrt{Q})$, so in order to get an upper bound for the expected regret of Algorithm 1, the key is to bound $\mathbb{E}[\Delta_q^{(k)}]$.

**Lemma 5.** *Under the setting of Theorem 1, we have*

$$\mathbb{E}[\Delta_q^{(k)}] \le \rho_k^2 \sigma^2 + (1-\rho_k)^2 \mathbb{E}[\Delta_q^{(k-1)}] + (1-\rho_k)^2 \frac{G}{(K-k+2)^2}$$

$$+ (1-\rho_k)^2 \left[ \frac{G}{\alpha_k(K-k+2)^2} + \alpha_k \mathbb{E}[\Delta_q^{(k-1)}] \right]$$

*where $\{\alpha_k\}$ is a sequence of positive parameters to be determined, $\sigma^2 = L_1^2 + \sigma_0^2$, and $G = (L_2 R + 2L_1)^2$.*

*Proof of Lemma 5.* By the definition of $d_q^{(k)}$, we have

$$\Delta_q^{(k)} = \|\nabla \bar{F}_{q,k-1}(x_q^{(k)}) - (1-\rho_k)d_q^{(k-1)} - \rho_k \tilde{\nabla} F_{t_{q,k}}(x_q^{(k)})\|^2$$

$$= \|\rho_k[\nabla \bar{F}_{q,k-1}(x_q^{(k)}) - \tilde{\nabla} F_{t_{q,k}}(x_q^{(k)})] + (1-\rho_k)[\nabla \bar{F}_{q,k-1}(x_q^{(k)}) - \nabla \bar{F}_{q,k-2}(x_q^{(k-1)})]$$

$$+ (1-\rho_k)[\nabla \bar{F}_{q,k-2}(x_q^{(k-1)}) - d_q^{(k-1)}]\|^2$$

$$= \rho_k^2 \|\nabla \bar{F}_{q,k-1}(x_q^{(k)}) - \tilde{\nabla} F_{t_{q,k}}(x_q^{(k)})\|^2 + (1-\rho_k)^2 \Delta_q^{(k-1)}$$

$$+ (1-\rho_k)^2 \|\nabla \bar{F}_{q,k-1}(x_q^{(k)}) - \nabla \bar{F}_{q,k-2}(x_q^{(k-1)})\|^2$$

$$+ 2\rho_k(1-\rho_k)\langle \nabla \bar{F}_{q,k-1}(x_q^{(k)}) - \tilde{\nabla} F_{t_{q,k}}(x_q^{(k)}), \nabla \bar{F}_{q,k-1}(x_q^{(k)}) - \nabla \bar{F}_{q,k-2}(x_q^{(k-1)})\rangle$$

$$+ 2\rho_k(1-\rho_k)\langle \nabla \bar{F}_{q,k-1}(x_q^{(k)}) - \tilde{\nabla} F_{t_{q,k}}(x_q^{(k)}), \nabla \bar{F}_{q,k-2}(x_q^{(k-1)}) - d_q^{(k-1)}\rangle$$

$$+ 2(1-\rho_k)^2 \langle \nabla \bar{F}_{q,k-1}(x_q^{(k)}) - \nabla \bar{F}_{q,k-2}(x_q^{(k-1)}), \nabla \bar{F}_{q,k-2}(x_q^{(k-1)}) - d_q^{(k-1)}\rangle. \tag{7}$$

For further analysis, we first denote $\mathcal{F}_{q,k}$ to be the $\sigma$-field generated by $t_{q,1}, t_{q,2}, \cdots, t_{q,k}$. Then by law of iterated expectations,

$$\mathbb{E}[\|\nabla \bar{F}_{q,k-1}(x_q^{(k)}) - \tilde{\nabla} F_{t_{q,k}}(x_q^{(k)})\|^2]$$

$$= \mathbb{E}[\mathbb{E}[\|\nabla \bar{F}_{q,k-1}(x_q^{(k)}) - \tilde{\nabla} F_{t_{q,k}}(x_q^{(k)})\|^2 | \mathcal{F}_{q,k-1}]]$$

$$= \mathbb{E}[\mathbb{E}[\|\nabla \bar{F}_{q,k-1}(x_q^{(k)}) - \nabla F_{t_{q,k}}(x_q^{(k)})\|^2 + \|\nabla F_{t_{q,k}}(x_q^{(k)}) - \tilde{\nabla} F_{t_{q,k}}(x_q^{(k)})\|^2$$

$$+ 2\langle \nabla \bar{F}_{q,k-1}(x_q^{(k)}) - \nabla F_{t_{q,k}}(x_q^{(k)}), \nabla F_{t_{q,k}}(x_q^{(k)}) - \tilde{\nabla} F_{t_{q,k}}(x_q^{(k)})\rangle | \mathcal{F}_{q,k-1}]]. \tag{8}$$

By Assumption 2, and $F_t$ is $L_1$-Lipschitz implies that $\sup_{x \in \mathcal{K}} \|\nabla F_t(x)\| \le L_1$, we have

$$\mathbb{E}[\mathbb{E}[\|\nabla \bar{F}_{q,k-1}(x_q^{(k)}) - \nabla F_{t_{q,k}}(x_q^{(k)})\|^2 | \mathcal{F}_{q,k-1}]] = \mathbb{E}[\text{Var}(\nabla F_{t_{q,k}}(x_q^{(k)}) | \mathcal{F}_{q,k-1})]$$

$$\le \mathbb{E}[\|\nabla F_{t_{q,k}}(x_q^{(k)})\|^2] \tag{9}$$

$$\le L_1^2.$$

By Assumption 3, we have

$$\mathbb{E}[\mathbb{E}[\|\nabla F_{t_{q,k}}(x_q^{(k)}) - \tilde{\nabla} F_{t_{q,k}}(x_q^{(k)})\|^2 | \mathcal{F}_{q,k-1}]] = \mathbb{E}[\|\nabla F_{t_{q,k}}(x_q^{(k)}) - \tilde{\nabla} F_{t_{q,k}}(x_q^{(k)})\|^2]$$

$$= \mathbb{E}[\mathbb{E}[\|\nabla F_{t_{q,k}}(x_q^{(k)}) - \tilde{\nabla} F_{t_{q,k}}(x_q^{(k)})\|^2 | \mathcal{F}_{q,k}]]$$

$$\le \sigma_0^2. \tag{10}$$

Moreover, we have

$$
\begin{aligned}
&\mathbb{E}[\mathbb{E}[\langle \nabla \bar{F}_{q,k-1}(x_q^{(k)}) - \nabla F_{t_{q,k}}(x_q^{(k)}), \nabla F_{t_{q,k}}(x_q^{(k)}) - \tilde{\nabla} F_{t_{q,k}}(x_q^{(k)})\rangle | \mathcal{F}_{q,k-1}]]\\
=&\mathbb{E}[\langle \nabla \bar{F}_{q,k-1}(x_q^{(k)}) - \nabla F_{t_{q,k}}(x_q^{(k)}), \nabla F_{t_{q,k}}(x_q^{(k)}) - \tilde{\nabla} F_{t_{q,k}}(x_q^{(k)})\rangle]\\
=&\mathbb{E}[\mathbb{E}[\langle \nabla \bar{F}_{q,k-1}(x_q^{(k)}) - \nabla F_{t_{q,k}}(x_q^{(k)}), \nabla F_{t_{q,k}}(x_q^{(k)}) - \tilde{\nabla} F_{t_{q,k}}(x_q^{(k)})\rangle | \mathcal{F}_{q,k}]]\\
=&\mathbb{E}[\langle \nabla \bar{F}_{q,k-1}(x_q^{(k)}) - \nabla F_{t_{q,k}}(x_q^{(k)}), \mathbb{E}[\nabla F_{t_{q,k}}(x_q^{(k)}) - \tilde{\nabla} F_{t_{q,k}}(x_q^{(k)}) | \mathcal{F}_{q,k}]\rangle]\\
=&0
\end{aligned}
\tag{11}
$$

where the last equation holds because $\tilde{\nabla} F_t$ is an unbiased estimator of $\nabla F_t$ for all $t$.

By Eqs. (8) to (11), we have

$$
\mathbb{E}[\|\nabla \bar{F}_{q,k-1}(x_q^{(k)}) - \tilde{\nabla} F_{t_{q,k}}(x_q^{(k)})\|^2] \leq L_1^2 + \sigma_0^2 \triangleq \sigma^2.
\tag{12}
$$

Similarly, by law of iterated expectations and the unbiasedness of $\tilde{\nabla} F_t$, we have

$$
\begin{aligned}
&\mathbb{E}[\langle \nabla \bar{F}_{q,k-1}(x_q^{(k)}) - \tilde{\nabla} F_{t_{q,k}}(x_q^{(k)}), \nabla \bar{F}_{q,k-1}(x_q^{(k)}) - \nabla \bar{F}_{q,k-2}(x_q^{(k-1)})\rangle]\\
=&\mathbb{E}[\mathbb{E}[\langle \nabla \bar{F}_{q,k-1}(x_q^{(k)}) - \tilde{\nabla} F_{t_{q,k}}(x_q^{(k)}), \nabla \bar{F}_{q,k-1}(x_q^{(k)}) - \nabla \bar{F}_{q,k-2}(x_q^{(k-1)})\rangle | \mathcal{F}_{q,k-1}]]\\
=&\mathbb{E}[\langle \mathbb{E}[\nabla \bar{F}_{q,k-1}(x_q^{(k)}) - \tilde{\nabla} F_{t_{q,k}}(x_q^{(k)}) | \mathcal{F}_{q,k-1}], \nabla \bar{F}_{q,k-1}(x_q^{(k)}) - \nabla \bar{F}_{q,k-2}(x_q^{(k-1)})\rangle]\\
=&0
\end{aligned}
\tag{13}
$$

and

$$
\begin{aligned}
&\mathbb{E}[\langle \nabla \bar{F}_{q,k-1}(x_q^{(k)}) - \tilde{\nabla} F_{t_{q,k}}(x_q^{(k)}), \nabla \bar{F}_{q,k-2}(x_q^{(k-1)}) - d_q^{(k-1)}\rangle]\\
=&\mathbb{E}[\mathbb{E}[\langle \nabla \bar{F}_{q,k-1}(x_q^{(k)}) - \tilde{\nabla} F_{t_{q,k}}(x_q^{(k)}), \nabla \bar{F}_{q,k-2}(x_q^{(k-1)}) - d_q^{(k-1)}\rangle | \mathcal{F}_{q,k-1}, d_q^{(k-1)}]]\\
=&\mathbb{E}[\langle \mathbb{E}[\nabla \bar{F}_{q,k-1}(x_q^{(k)}) - \tilde{\nabla} F_{t_{q,k}}(x_q^{(k)}) | \mathcal{F}_{q,k-1}, d_q^{(k-1)}], \nabla \bar{F}_{q,k-2}(x_q^{(k-1)}) - d_q^{(k-1)}\rangle]\\
=&0.
\end{aligned}
\tag{14}
$$

Also, by Young's Inequality, we have

$$
\begin{aligned}
\langle \nabla \bar{F}_{q,k-1}(x_q^{(k)}) &- \nabla \bar{F}_{q,k-2}(x_q^{(k-1)}), \nabla \bar{F}_{q,k-2}(x_q^{(k-1)}) - d_q^{(k-1)}\rangle\\
&\leq \frac{1}{2\alpha_k}\|\nabla \bar{F}_{q,k-1}(x_q^{(k)}) - \nabla \bar{F}_{q,k-2}(x_q^{(k-1)})\|^2 + \frac{\alpha_k}{2}\Delta_q^{(k-1)}.
\end{aligned}
\tag{15}
$$

Now we turn to bound $\|\nabla \bar{F}_{q,k-1}(x_q^{(k)}) - \nabla \bar{F}_{q,k-2}(x_q^{(k-1)})\|^2 \triangleq z_{q,k}^2$. In fact, we have

$$
\begin{aligned}
\mathbb{E}[z_{q,k}^2] &= \mathbb{E}[\mathbb{E}[\|\nabla \bar{F}_{q,k-1}(x_q^{(k)}) - \nabla \bar{F}_{q,k-2}(x_q^{(k-1)})\|^2 | \mathcal{F}_{q,k-2}]]\\
&= \mathbb{E}[\mathbb{E}[\|\frac{\sum_{i=k}^{K} \nabla F_{t_{q,i}}(x_q^{(k)})}{K-k+1} - \frac{\sum_{i=k-1}^{K} \nabla F_{t_{q,i}}(x_q^{(k-1)})}{K-k+2}\|^2 | \mathcal{F}_{q,k-2}]]\\
&= \mathbb{E}[\mathbb{E}[\|\frac{\sum_{i=k}^{K} \nabla F_{t_{q,i}}(x_q^{(k)}) - \nabla F_{t_{q,i}}(x_q^{(k-1)})}{K-k+2} + \frac{\sum_{i=k}^{K} \nabla F_{t_{q,i}}(x_q^{(k)})}{(K-k+1)(K-k+2)}\\
&\qquad - \frac{\nabla F_{t_{q,k-1}}(x_q^{(k-1)})}{K-k+2}\|^2 | \mathcal{F}_{q,k-2}]]\\
&\leq \mathbb{E}[\mathbb{E}[(\sum_{i=k}^{K}\|\frac{\nabla F_{t_{q,i}}(x_q^{(k)}) - \nabla F_{t_{q,i}}(x_q^{(k-1)})}{K-k+2}\| + \sum_{i=k}^{K}\|\frac{\nabla F_{t_{q,i}}(x_q^{(k)})}{(K-k+1)(K-k+2)}\|\\
&\qquad + \|\frac{\nabla F_{t_{q,k-1}}(x_q^{(k-1)})}{K-k+2}\|)^2 | \mathcal{F}_{q,k-2}]].
\end{aligned}
$$

where the inequality comes from the Triangle Inequality of norms.

Recall the update rule where $x_q^{(k)} = x_q^{(k-1)} + \frac{1}{K} v_q^{(k-1)}$ and the assumption that $F_t$ is $L_2$-smooth, we have

$$\|\nabla F_{t_{q,i}}(x_q^{(k)}) - \nabla F_{t_{q,i}}(x_q^{(k-1)})\| \leq L_2 \frac{\|v_q^{(k)}\|}{K} = \frac{L_2 R}{K}.$$

Also by Assumption 2, $\|\nabla F_{t_{q,i}}(x_q^{(k-1)})\| \leq L_1$. Therefore, we have

$$\mathbb{E}[z_{q,k}^2] \leq [(K-k+1)\frac{L_2 R}{K}\frac{1}{K-k+2} + \frac{L_1}{K-k+2} + (K-k+1)\frac{L_1}{(K-k+1)(K-k+2)}]^2$$

$$\leq \left(\frac{L_2 R + 2L_1}{K-k+2}\right)^2$$

$$\triangleq \frac{G}{(K-k+2)^2}.$$

(16)

Combining Eqs. (7), (12), (13), (14), (15) and (16), we have

$$\mathbb{E}[\Delta_q^{(k)}] \leq \rho_k^2 \sigma^2 + (1-\rho_k)^2 \mathbb{E}[\Delta_q^{(k-1)}] + (1-\rho_k)^2 \frac{G}{(K-k+2)^2}$$

$$+ (1-\rho_k)^2 \left[\frac{G}{\alpha_k(K-k+2)^2} + \alpha_k \mathbb{E}[\Delta_q^{(k-1)}]\right].$$

$\square$

Applying Lemma 5 and setting $\alpha_k = \frac{\rho_k}{2}, \forall k \in 1, 2, \cdots, K$, we have

$$\mathbb{E}[\Delta_q^{(k)}] \leq \rho_k^2 \sigma^2 + \frac{G}{(K-k+2)^2}(1-\rho_k)^2\left(1+\frac{2}{\rho_k}\right) + \mathbb{E}[\Delta_q^{(k-1)}](1-\rho_k)^2\left(1+\frac{\rho_k}{2}\right).$$

Note that if $0 < \rho_k \leq 1$, then we have

$$(1-\rho_k)^2\left(1+\frac{2}{\rho_k}\right) \leq \left(1+\frac{2}{\rho_k}\right)$$

and

$$(1-\rho_k)^2\left(1+\frac{\rho_k}{2}\right) \leq (1-\rho_k).$$

So in this case, we have

$$\mathbb{E}[\Delta_q^{(k)}] \leq \rho_k^2 \sigma^2 + \frac{G}{(K-k+2)^2}\left(1+\frac{2}{\rho_k}\right) + \mathbb{E}[\Delta_q^{(k-1)}](1-\rho_k).$$

(17)

**Lemma 6.** *Under the setting of Theorem 1, we have*

$$\mathbb{E}[\Delta_q^{(k)}] \leq \begin{cases} \frac{N}{(k+4)^{2/3}}, & \text{when } 1 \leq k \leq \frac{K}{2}. \\ \frac{N}{(K-k+1)^{2/3}}, & \text{when } \frac{K}{2}+1 \leq k \leq K. \end{cases}$$

*where $N = \max\{5^{2/3}(L_1 + M_0)^2, 4\sigma^2 + 32G, 2.25\sigma^2 + 7G/3\}$.*

*Proof of Lemma 6.* When $1 \leq k \leq \frac{K}{2} + 1$, since $\rho_k = \frac{2}{(k+3)^{2/3}}$, we have $0 < \rho_k \leq 1$, and by Eq. (17)

$$
\begin{aligned}
\mathbb{E}[\Delta_q^{(k)}] &\leq \frac{4\sigma^2}{(k+3)^{4/3}} + \frac{G}{k^2}[1 + (k+3)^{2/3}] + \mathbb{E}[\Delta_q^{(k-1)}]\left(1 - \frac{2}{(k+3)^{2/3}}\right) \\
&= \frac{4\sigma^2}{(k+3)^{4/3}} + \frac{G}{(k+3)^2}\left(\frac{k+3}{k}\right)^2[1 + (k+3)^{2/3}] + \mathbb{E}[\Delta_q^{(k-1)}]\left(1 - \frac{2}{(k+3)^{2/3}}\right) \\
&\leq \frac{4\sigma^2}{(k+3)^{4/3}} + \frac{G(1+3)^2}{(k+3)^2}[1 + (k+3)^{2/3}] + \mathbb{E}[\Delta_q^{(k-1)}]\left(1 - \frac{2}{(k+3)^{2/3}}\right) \\
&\leq \frac{4\sigma^2}{(k+3)^{4/3}} + \frac{16G}{(k+3)^{4/3}} + \frac{16G}{(k+3)^{4/3}} + \mathbb{E}[\Delta_q^{(k-1)}]\left(1 - \frac{2}{(k+3)^{2/3}}\right) \\
&= \frac{4\sigma^2 + 32G}{(k+3)^{4/3}} + \mathbb{E}[\Delta_q^{(k-1)}]\left(1 - \frac{2}{(k+3)^{2/3}}\right) \\
&\triangleq \frac{N_0}{(k+3)^{4/3}} + \mathbb{E}[\Delta_q^{(k-1)}]\left(1 - \frac{2}{(k+3)^{2/3}}\right).
\end{aligned}
$$

Recall that $\Delta_q^{(k)} = \|\nabla \bar{F}_{q,k-1}(x_q^{(k)}) - d_q^{(k)}\|^2$, and thus

$$
\begin{aligned}
\Delta_q^{(1)} &= \|\nabla \bar{F}_{q,0}(0) - d_q^{(1)}\|^2 \\
&= \left\|\frac{\sum_{i=1}^{K} \nabla F_{t_{q,i}}(0)}{K} - \frac{2}{(1+3)^{2/3}}\tilde{\nabla}F_{q,1}(0)\right\|^2 \\
&\leq \left(\sum_{i=1}^{K}\left\|\frac{\nabla F_{t_{q,i}}(0)}{K}\right\| + \left\|\frac{2}{4^{2/3}}\tilde{\nabla}F_{q,1}(0)\right\|\right)^2 \\
&\leq \left(K\frac{L_1}{K} + M_0\right)^2 \\
&= (L_1 + M_0)^2.
\end{aligned}
$$

Set $N_1 = \max\{5^{2/3}(L_1 + M_0)^2, N_0\}$, then we claim that $\mathbb{E}[\Delta_q^{(k)}] \leq \frac{N_1}{(k+4)^{2/3}}$ for any $k$ satisfying $1 \leq k \leq \frac{K}{2} + 1$. We prove it by induction. It holds for $k = 1$ because of the definition of $N_1$. Assume it holds for $k-1$, *i.e.*, $\mathbb{E}[\Delta_q^{(k-1)}] \leq \frac{N_1}{(k+3)^{2/3}}$, then

$$
\begin{aligned}
\mathbb{E}[\Delta_q^{(k)}] &\leq \frac{N_1}{(k+3)^{4/3}} + \mathbb{E}[\Delta_q^{(k-1)}]\left(1 - \frac{2}{(k+3)^{2/3}}\right) \\
&\leq \frac{N_1}{(k+3)^{4/3}} + \frac{N_1}{(k+3)^{2/3}}\left(1 - \frac{2}{(k+3)^{2/3}}\right) \\
&= \frac{N_1[(k+3)^{2/3} - 1]}{(k+3)^{4/3}}.
\end{aligned}
$$

Since $(k+4)^2 = k^2 + 8k + 16 \leq k^2 + 6k + 9 + 1 + 3(k+3) \leq k^2 + 6k + 9 + 1 + 3(k+3)^{4/3} + 3(k+3)^{2/3} = [(k+3)^{2/3} + 1]^3$, by taking the cube roots of both sides, we have $(k+4)^{2/3} \leq (k+3)^{2/3} + 1$, which implies that $[(k+3)^{2/3} - 1](k+4)^{2/3} \leq [(k+3)^{2/3} - 1][(k+3)^{2/3} + 1] \leq (k+3)^{4/3}$, *i.e.*, $\frac{(k+3)^{2/3}-1}{(k+3)^{4/3}} \leq \frac{1}{(k+4)^{2/3}}$. So we have $\mathbb{E}[\Delta_q^{(k)}] \leq \frac{N_1}{(k+4)^{2/3}}$. By induction, we have

$$
\mathbb{E}[\Delta_q^{(k)}] \leq \frac{N_1}{(k+4)^{2/3}}, \forall k \in [\frac{K}{2} + 1]. \tag{18}
$$

Now we turn to consider the case where $\frac{K}{2} + 2 \leq k \leq K$. Here we set $\rho_k = \frac{1.5}{(K-k+2)^{2/3}}$, note that $0 < \rho_k \leq \frac{1.5}{2^{2/3}} < 1$, then we have

$$
\begin{aligned}
\mathbb{E}[\Delta_q^{(k)}] &\leq \frac{2.25\sigma^2}{(K-k+2)^{4/3}} + \frac{G}{(K-k+2)^2}\left[1 + \frac{4}{3}(K-k+2)^{2/3}\right] \\
&\quad + \mathbb{E}[\Delta_q^{(k-1)}]\left[1 - \frac{1.5}{(K-k+2)^{2/3}}\right] \\
&\leq \frac{2.25\sigma^2}{(K-k+2)^{4/3}} + \frac{G}{(K-k+2)^{4/3}} + \frac{4}{3}\frac{G}{(K-k+2)^{4/3}} \\
&\quad + \mathbb{E}[\Delta_q^{(k-1)}]\left[1 - \frac{1.5}{(K-k+2)^{2/3}}\right] \\
&= \frac{2.25\sigma^2 + 7G/3}{(K-k+2)^{4/3}} + \mathbb{E}[\Delta_q^{(k-1)}]\left[1 - \frac{1.5}{(K-k+2)^{2/3}}\right] \\
&\triangleq \frac{N_2}{(K-k+2)^{4/3}} + \mathbb{E}[\Delta_q^{(k-1)}]\left[1 - \frac{1.5}{(K-k+2)^{2/3}}\right].
\end{aligned}
$$

Define $N = \max\{N_1, N_2\}$, then we claim that $\mathbb{E}[\Delta_q^{(k)}] \leq \frac{N}{(K-k+1)^{2/3}}$, for any $k$ satisfying $\frac{K}{2} + 1 \leq k \leq K$, we will prove it by induction. When $k = \frac{K}{2} + 1$, by Eq. (18), we have

$$
\mathbb{E}[\Delta_q^{(K/2+1)}] \leq \frac{N_1}{(K/2+1+4)^{2/3}} \leq \frac{N}{(K/2)^{2/3}} = \frac{N}{(K-(K/2+1)+1)^{2/3}}.
$$

When it holds for $k - 1$, i.e., $\mathbb{E}[\Delta_q^{(k-1)}] \leq \frac{N}{(K-k+2)^{2/3}}$, we have

$$
\begin{aligned}
\mathbb{E}[\Delta_q^{(k)}] &\leq \frac{N}{(K-k+2)^{4/3}} + \frac{N}{(K-k+2)^{2/3}}\frac{(K-k+2)^{2/3} - 1.5}{(K-k+2)^{2/3}} \\
&= \frac{N[(K-k+2)^{2/3} - 0.5]}{(K-k+2)^{4/3}}.
\end{aligned}
$$

Since $[(K-k+2)^{2/3} - 0.5](K-k+1)^{2/3} \leq [(K-k+2)^{2/3} - 0.5][(K-k+2)^{2/3} + 0.5] \leq (K-k+2)^{4/3}$, i.e., $\frac{(K-k+2)^{2/3}-0.5}{(K-k+2)^{4/3}} \leq \frac{1}{(K-k+1)^{2/3}}$, so we have $\mathbb{E}[\Delta_q^{(k)}] \leq \frac{N}{(K-k+1)^{2/3}}$. By induction, we have

$$
\mathbb{E}[\Delta_q^{(k)}] \leq \frac{N}{(K-k+1)^{2/3}}, \forall k \in \{K/2+1, K/2+2, \cdots, K\}.
$$

Since $N_1 \leq N$, by Eq. (18), we also have

$$
\mathbb{E}[\Delta_q^{(k)}] \leq \frac{N}{(k+4)^{2/3}}, \forall k \in [\frac{K}{2}+1].
$$

$\square$

Recall that in Eq. (6), we have

$$
\mathbb{E}[\mathcal{R}_T] \leq \sum_{q=1}^{Q}\sum_{k=1}^{K}\frac{1}{2\beta^{(k)}}\mathbb{E}[\Delta_q^{(k)}] + \frac{D^2}{2}Q\sum_{k=1}^{K}\beta^{(k)} + KR_Q^{\mathcal{E}} + \frac{L_2 D^2}{2}Q.
$$

So if we set

$$
\beta^{(k)} = \begin{cases} (k+4)^{-1/3}, & \text{when } 1 \leq k \leq \frac{K}{2}; \\ (K-k+1)^{-1/3}, & \text{when } \frac{K}{2}+1 \leq k \leq K; \end{cases}
$$

then by Lemma 6, we have

$$
\sum_{k=1}^{K/2}\frac{\mathbb{E}[\Delta_q^{(k)}]}{\beta^{(k)}} \leq \sum_{k=1}^{K/2}\frac{N}{(k+4)^{1/3}} \leq \sum_{k=1}^{K/2}\frac{N}{k^{1/3}} \leq \int_{0}^{K/2}\frac{N}{x^{1/3}}\mathrm{d}x = \frac{3N}{2}\left(\frac{K}{2}\right)^{2/3} \leq NK^{2/3},
$$

and

$$\sum_{k=K/2+1}^{K} \frac{\mathbb{E}[\Delta_q^{(k)}]}{\beta^{(k)}} \leq \sum_{k=K/2+1}^{K} \frac{N}{(K-k+1)^{1/3}} = \sum_{i=1}^{K/2} \frac{N}{i^{1/3}} \leq NK^{2/3}.$$

Similarly, we have

$$\sum_{k=1}^{K/2} \beta^{(k)} = \sum_{k=1}^{K/2} \frac{1}{(k+4)^{1/3}} \leq K^{2/3}$$

and

$$\sum_{k=K/2+1}^{K} \beta^{(k)} = \sum_{k=K/2+1}^{K} \frac{1}{(K-k+1)^{1/3}} \leq K^{2/3}.$$

Therefore, we have

$$\mathbb{E}[\mathcal{R}_T] \leq \sum_{q=1}^{Q} NK^{2/3} + \frac{D^2}{2}Q \cdot 2K^{2/3} + K\mathcal{R}_Q^{\mathcal{E}} + \frac{L_2 D^2}{2}Q$$

$$= (N + D^2)QK^{2/3} + K\mathcal{R}_Q^{\mathcal{E}} + \frac{L_2 D^2}{2}Q.$$

Set $Q = T^{2/5}$, $K = T^{3/5}$, and recall that $\mathcal{R}_Q^{\mathcal{E}} \leq C\sqrt{Q} = CT^{1/5}$, we have

$$\mathbb{E}[\mathcal{R}_T] \leq (N + C + D^2)T^{4/5} + \frac{L_2 D^2}{2}T^{2/5}.$$

$\square$

## C  Properties of Smoothed Functions

**Lemma 7.** *If $F$ is monotone, continuous DR-submodular, $L_1$-Lipschitz, and $L_2$-smooth, then so is $\hat{F}_\delta$, and for all $x$ we have $|\hat{F}_\delta(x) - F(x)| \leq L_1\delta$.*

*Proof.* By Lemmas 1 and 2 of [19], we conclude that $\hat{F}_\delta$ is also monotone continuous DR-submodular, $L_1$-Lipschitz and it holds that

$$|\hat{F}_\delta(x) - F(x)| \leq L_1\delta.$$

For any $x, y$ in the domain of $\hat{F}_\delta$, we have

$$\begin{aligned}
\|\nabla \hat{F}_\delta(x) - \nabla \hat{F}_\delta(y)\| &= \|\nabla\mathbb{E}[F(x + \delta v)] - \nabla\mathbb{E}[F(y + \delta v)]\| \\
&= \|\mathbb{E}[\nabla F(x + \delta v)] - \mathbb{E}[\nabla F(y + \delta v)]\| \\
&= \|\mathbb{E}[\nabla F(x + \delta v) - \nabla F(y + \delta v)]\| \\
&\leq \mathbb{E}[\|\nabla F(x + \delta v) - \nabla F(y + \delta v)\|] \\
&\leq \mathbb{E}[L_2\|x - y\|] \\
&= L_2\|x - y\|.
\end{aligned}$$

So $\hat{F}_\delta$ is also $L_2$-smooth.

$\square$

## D  Construction of $\delta$-Interior

Fig. 1 is the illustrations of $\delta$-interior and the construction method as discussed in Lemma 1.

Now we turn to prove Lemma 1. We first show the following auxiliary lemma.

**Lemma 8.** *Consider a ball centered at the origin $o$. If point $a$ resides on the sphere but not in the non-negative orthant, there must exist a point $b$ on the sphere such that all the components of $\overrightarrow{ab}$ are positive and all the components of $\overrightarrow{ob}$ are non-negative.*

(a) Example of $\delta$-interior    (b) Construction of $\delta$-interior

Figure 1: $\delta$-interior

*Proof of Lemma 8.* Without loss of generality, we assume the Cartesian coordinates of $a$ are $(-\epsilon_1, -\epsilon_2, \cdots, -\epsilon_k, \epsilon_{k+1}, \cdots, \epsilon_d)$, where $\epsilon_i > 0, \forall i \in [k], \epsilon_j \geq 0, \forall j \in \{k+1, \cdots, d\}$, and $k \in [d]$. In order to find a point $b$, we first define the symmetric point $b' = (\epsilon_1, \epsilon_2, \cdots, \epsilon_k, \epsilon_{k+1}, \cdots, \epsilon_d)$.

If $k = d$, we can set $b = b'$, then $b$ is on the sphere, $b_i - a_i = 2\epsilon_i > 0$, and $b_i = \epsilon_i > 0, \forall i \in [d]$.

If $k < d$, we can add some perturbations on $b'$. Let $\epsilon = \min\{\epsilon_1, \epsilon_2, \cdots, \epsilon_k\} > 0, A = \frac{2\epsilon \sum_{i=1}^{k} \epsilon_i - k\epsilon^2}{d-k} > 0$, and set $b = b' + (-\epsilon, -\epsilon, \cdots, -\epsilon, \sqrt{A + \epsilon_{k+1}^2} - \epsilon_{k+1}, \cdots, \sqrt{A + \epsilon_d^2} - \epsilon_d) = (\epsilon_1 - \epsilon, \epsilon_2 - \epsilon, \cdots, \epsilon_k - \epsilon, \sqrt{A + \epsilon_{k+1}^2}, \cdots, \sqrt{A + \epsilon_d^2})$. Note that $|ob|^2 = \sum_{i=1}^{k}(\epsilon_i - \epsilon)^2 + \sum_{j=k+1}^{d}(A + \epsilon_j^2) = \sum_{i=1}^{k}\epsilon_i^2 - 2\epsilon\sum_{i=1}^{k}\epsilon_i + k\epsilon^2 + 2\epsilon\sum_{i=1}^{k}\epsilon_i - k\epsilon^2 + \sum_{j=k+1}^{d}\epsilon_j^2 = \sum_{l=1}^{d}\epsilon_l^2 = |oa|^2$, so $b$ is also on the sphere. Moreover, $b_i - a_i = 2\epsilon_i - \epsilon > 0, \forall i \in [k], b_j - a_j = \sqrt{A + \epsilon_j^2} - \epsilon_j > 0, \forall j \in \{k+1, \cdots, d\}$, and $b_i = \epsilon_i - \epsilon \geq 0, \forall i \in [k], b_j = \sqrt{A + \epsilon_j^2} > 0, \forall j \in \{k+1, \cdots, d\}$.

Therefore, all the scalar components of $\overrightarrow{ab}$ are positive, and all the scalar components of $\overrightarrow{ob}$ are non-negative. $\qquad\square$

Figure 2: Illustrations for Proof of Lemma 1

*Proof of Lemma 1.* Since $\mathcal{K}$ is convex, compact, and down-closed, and only shrinkage and translation are involved, so $\mathcal{K}'$ is also convex, compact, and down-closed. In order to prove that $\mathcal{K}'$ is a $\delta$-interior of $\mathcal{K}$, note that thanks to the $\delta\mathbf{1}$ translation, the distance between $\mathcal{K}'$ and the face which contains 0 (*i.e.*, the set $\partial^0\mathcal{K} = \{x \in \partial\mathcal{K} | \exists i \in [d] \text{ such that } x_i = 0\}$), is no less than $\delta$. In other words, for every $a^* \in \mathcal{K}'$, we have $\inf_{x \in \partial^0\mathcal{K}} d(x, a^*) \geq \delta$.

So we only need to consider the remaining points on $\partial\mathcal{K}$, which we denote as $\partial^*\mathcal{K} = \partial\mathcal{K} \setminus \partial^0\mathcal{K} = \{x \in \partial\mathcal{K} | \forall i \in [d], x_i > 0\}$. We also denote the closure of $\partial^*\mathcal{K}$ as $\text{cl}(\partial^*\mathcal{K})$, which is a subset of $\partial\mathcal{K}$.

Since for every point $a^* \in \mathcal{K}'$, there is a point $a' = a^* - \delta \mathbf{1} \in \mathcal{K}_\alpha$, and $|a'a^*| = \sqrt{d}\delta$, we can first analyze $\inf_{s \in \partial^* \mathcal{K}} d(s, a')$, and then upper bound $\inf_{s \in \partial^* \mathcal{K}} d(s, a^*)$ by triangle inequality.

For any point $a' \in \mathcal{K}_\alpha$, suppose the point $a \in \mathrm{cl}(\partial^* \mathcal{K})$ satisfies $|aa'| = \inf_{x \in \partial^* \mathcal{K}} d(x, a')$ (Fig. 2a). We claim that all the scalar components of the vector $\overrightarrow{a'a}$ are non-negative. We will prove it by contradiction. Consider a ball with $a'$ as the center and $|a'a|$ as the radius. If we regard $a'$ as the origin $o$, then the assumption that $\overrightarrow{a'a}$ has negative scalar component is equivalent to that $a$ is not in the non-negative orthant.

By Lemma 8, there exists a point $b$, such that $|a'b| = |a'a|$, all the scalar components of $\overrightarrow{ab}$ are positive, and all the scalar components of $\overrightarrow{a'b}$ are non-negative (Fig. 2b). Then we claim $b \in \mathcal{K}$, which will be also proved by contradiction. If $b \notin \mathcal{K}$, since $a \in \mathrm{cl}(\partial^* \mathcal{K})$ implies $a_i \geq 0, \forall i$, the fact that all the scalar components of $\overrightarrow{ab}$ are positive implies $b_i > 0, \forall i$.

Since $a' \in \mathcal{K}_\alpha$, there must be a point $c \neq a'$ in the line segment $\overline{a'b}$ such that $c \in \partial \mathcal{K}$. To prove it, note that $a' \in \mathcal{K}_\alpha \implies a' \in (1-\alpha)\mathcal{K}$, and $(\sqrt{d}+1)\delta B^d_{\geq 0} = \alpha r B^d_{\geq 0} \subseteq \alpha \mathcal{K}$. So, $a' + (\sqrt{d}+1)\delta B^d_{\geq 0} \subseteq (1-\alpha)\mathcal{K} + \alpha \mathcal{K} = \mathcal{K}$ by the convexity of $\mathcal{K}$. On the other hand, since all the scalar components of $\overrightarrow{a'b}$ are non-negative, the intersection between the line segment $\overline{a'b}$ and the set $a' + (\sqrt{d}+1)\delta B^d_{\geq 0}$ must contains point other than $a'$. We denote this point as $c'$, then $c' \in a' + (\sqrt{d}+1)\delta B^d_{\geq 0} \subseteq \mathcal{K}$. By the convexity of $\mathcal{K}$, the continuity of the line segment $\overline{a'b}$, and the assumption that $b \notin \mathcal{K}$, there must be a point $c \neq a'$ in $\overline{a'b}$ such that $c \in \partial \mathcal{K}$.

Then $c \neq a', a'_i \geq 0, b_i > 0, c \in \overline{a'b}$ imply that $c_i > 0, \forall i$, thus $c \in \partial^* \mathcal{K}$. Moreover, since we assume $b \notin \mathcal{K}$, we have $|a'c| < |a'b| = |a'a|$, which is contradictory with the assumption that $|a'a| = \inf_{x \in \partial^* \mathcal{K}} d(x, a')$.

So we must have $b \in \mathcal{K}$. Since the scalar components of $\overrightarrow{ab}$ all all positive, and $\mathcal{K}$ is down-closed ($0 \leq x \leq y, y \in \mathcal{K} \implies x \in \mathcal{K}$), we conclude that $a$ is an interior point of $\mathcal{K}$ (Fig. 2c), which is contradictory to the assumption that $a \in \mathrm{cl}(\partial^* \mathcal{K})$. So we have proved that all the scalar components of the vector $\overrightarrow{a'a}$ are non-negative.

Then we proceed to show $|a'a| \geq (\sqrt{d}+1)\delta$. Let $v$ be the vector $\frac{(\sqrt{d}+1)\delta}{|a'a|}\overrightarrow{a'a}$, and $p$ be the point such that $\overrightarrow{a'p} = v$ (Fig. 2a). Then $|v| = (\sqrt{d}+1)\delta$ and all the scalar components of $v$ are non-negative, $i.e.$, $v \in (\sqrt{d}+1)\delta B^d_{\geq 0} = \alpha r B^d_{\geq 0} \subseteq \alpha \mathcal{K}$. We also have $a' \in \mathcal{K}_\alpha = (1-\alpha)\mathcal{K}$, thus $p \in (1-\alpha)\mathcal{K} + \alpha \mathcal{K} = \mathcal{K}$ by the convexity of $\mathcal{K}$. Since $a \in \mathrm{cl}(\partial^* \mathcal{K})$, we have $|a'a| \geq |a'p| = |v| = (\sqrt{d}+1)\delta$.

Let $a^* = a' + \delta \mathbf{1}$ be the translated point of $a'$. Then for any point $s \in \partial^* \mathcal{K}$, by triangle inequality, we have $|a^*s| \geq |a's| - |a'a^*| \geq |a'a| - |a'a^*| \geq (\sqrt{d}+1)\delta - \sqrt{d}\delta = \delta$. So $\inf_{x \in \partial^* \mathcal{K}} d(x, a^*) \geq \delta$. Since $a'$ can be arbitrary point in $\mathcal{K}_\alpha$, the inequality holds for every point $a^* \in \mathcal{K}'$. Recall that we have proved that for every $a^* \in \mathcal{K}'$, $\inf_{x \in \partial^0 \mathcal{K}} d(x, a^*) \geq \delta$, where $\partial^0 \mathcal{K} = \{x \in \partial \mathcal{K} | \exists i \in [d] \text{ such that } x_i = 0\} = \partial \mathcal{K} \setminus \partial^* \mathcal{K}$. Therefore, we conclude that for every point $a^* \in \mathcal{K}'$, $\inf_{x \in \partial \mathcal{K}} d(x, a^*) \geq \delta$.

So we only need to prove $\mathcal{K}' \subseteq \mathcal{K}$. For every $a^* \in \mathcal{K}'$, since $a' = a^* - \delta \mathbf{1} \in \mathcal{K}_\alpha$, there must be a positive $\beta$, such that $\tilde{a} = a' + \beta \mathbf{1} \in \partial^* \mathcal{K}$ (Fig. 2d). We have shown that $\inf_{x \in \partial^* \mathcal{K}} d(x, a') \geq (\sqrt{d}+1)\delta$, so $\beta \geq \frac{\sqrt{d}+1}{\sqrt{d}}\delta > \delta$. So $a^* = a' + \delta \mathbf{1}$ must be in the segment of $\overline{a'\tilde{a}}$. Then we have $a^* \in \mathcal{K}$, by the fact that $a', \tilde{a} \in \mathcal{K}$, and the convexity of $\mathcal{K}$. Therefore, $\mathcal{K}' \subseteq \mathcal{K}$, and thus $\mathcal{K}'$ is a $\delta$-interior of $\mathcal{K}$.

Now we turn to analyze $d(\mathcal{K}, \mathcal{K}')$. For any point $x \in \mathcal{K}$, we define $x' = (1-\alpha)x \in \mathcal{K}_\alpha$, and have $|xx'| = \alpha|ox| \leq \alpha R$. Let $x^* = x' + \delta \mathbf{1} \in \mathcal{K}'$, then $|xx^*| \leq |xx'| + |x'x^*| \leq \alpha R + \sqrt{d}\delta = [\sqrt{d}(\frac{R}{r}+1) + \frac{R}{r}]\delta$. Thus $d(\mathcal{K}, \mathcal{K}') \leq [\sqrt{d}(\frac{R}{r}+1) + \frac{R}{r}]\delta$. $\qquad \square$

# E  Analysis of Algorithm 2

## E.1  General Constraint Set

We first state a necessary assumption on the $\delta$-interior $\mathcal{K}'$.

**Assumption 8.** *For sufficiently small $\delta > 0$, the $\delta$-interior $\mathcal{K}'$ is convex and compact, and has lower bound $\underline{u}$ such that $\forall x \in \mathcal{K}', x \geq \underline{u}$. We also assume that the discrepancy satisfies $d(\mathcal{K}, \mathcal{K}') \leq c_1 \delta^\gamma$, where $c_1, \gamma > 0$.*

Note that we have $\sup_{x,y \in \mathcal{K}'} \|x - y\| \leq D, \sup_{x \in \mathcal{K}'} \|x - \underline{u}\| \leq R$, where $D, R$ are the diameter and radius of $\mathcal{K}$. In other words, the bounds for $\mathcal{K}$ also hold for $\mathcal{K}'$.

Also, if the constraint set $\mathcal{K}$ satisfies Assumption 1 and is down-closed, Lemma 1 shows that one can construct a $\delta$-interior $\mathcal{K}'$ that obeys Assumption 8.

Now with the assumption on the reward functions $F_t$ (Assumptions 2 and 6), and those on $\mathcal{K}$ and $\mathcal{K}'$ (Assumptions 1 and 8), we show Algorithm 2 achieves a sublinear $(1 - 1/e)$-regret bound of $O(T^{\frac{3+5\min\{1,\gamma\}}{3+6\min\{1,\gamma\}}})$.

**Theorem 4.** *Under Assumptions 1, 2, 4, 6 and 8, if we set $\delta = c_2 T^{-\frac{1}{3+6\min\{1,\gamma\}}}, Q = T^{\frac{2\min\{1,\gamma\}}{3+6\min\{1,\gamma\}}}, L = T^{\frac{3+4\min\{1,\gamma\}}{3+6\min\{1,\gamma\}}}, K = T^{\frac{1+\min\{1,\gamma\}}{1+2\min\{1,\gamma\}}}, \eta_k = \frac{1}{K}, \rho_k = \frac{2}{(k+2)^{2/3}}$, where $c_2 > 0$ is a constant such that $\delta$ is sufficiently small as required by Assumption 8, then the expected $(1 - 1/e)$-regret of Algorithm 2 is at most*

$$\mathbb{E}[\mathcal{R}_T] \leq \left[(1 - 1/e)c_1 c_2^\gamma L_1 + (2 - 1/e)c_2 L_1 + 2M_1 + \frac{3 \cdot 4^{1/6} d^2 M_1^2}{c_2} + \frac{3D^2}{4c_2} + C\right] T^{\frac{3+5\min\{1,\gamma\}}{3+6\min\{1,\gamma\}}}$$
$$+ \frac{3c_2[2L_1^2 + (3L_2 R + 2L_1)^2]}{4^{1/3}} T^{\frac{1+5\min\{1,\gamma\}}{3+6\min\{1,\gamma\}}} + \frac{L_2 D^2}{2} T^{\frac{\min\{1,\gamma\}}{1+2\min\{1,\gamma\}}}.$$

*Proof of Theorem 4.* Since $x_q^{(1)} = \underline{u}$ and $\eta_k = 1/K$, $x_q^{(k)}$ is actually a convex combination of $\underline{u}, v_q^{(1)}, v_q^{(2)}, \cdots, v_q^{(k-1)}$. Then $\underline{u} \in \mathcal{K}', v_q^{(i)} \in \mathcal{K}', \forall i \in [K]$ implies $x_q^{(k)} \in \mathcal{K}', \forall k \in [K+1]$. So for $k \in [K], y_{t_{q,k}} = x_q^{(k)} + \delta u_{q,k} \in \mathcal{K}$; for $t \in \{(q-1)L + 1, \cdots, qL\} \setminus \{t_{q,1}, \cdots, t_{q,K}\}$, $y_t = x_q = x_q^{(K+1)} \in \mathcal{K}' \subseteq \mathcal{K}$. In other words, all the points that we play fall on the constraint set $\mathcal{K}$.

We also note that as discussed before, the regret bound for online linear oracle, $\mathcal{R}_t^{\mathcal{E}} \leq C\sqrt{t}$ can be achieved by algorithms such as Online Gradient Descent.

Then we define

$$\hat{F}_{t,\delta}(x) = \mathbb{E}_{v \sim B^d}[F_t(x + \delta v)]$$

as the $\delta$-smoothed version of $F_t$. We omit the $\delta$ in the subscript for simplicity in the rest of the proof. Since $F_t$ is $L_1$-Lipschitz, by Lemma 7 in Appendix C, we have

$$|\hat{F}_t(x) - F_t(x)| \leq L_1 \delta.$$

Therefore, if we define $x^* = \arg\max_{x\in\mathcal{K}}\sum_{t=1}^{T}F_t(x)$, $x_\delta^* = \arg\max_{x\in\mathcal{K}'}\sum_{t=1}^{T}F_t(x)$, the $(1-1/e)$-regret with horizon $T$ is

$$\mathcal{R}_T = \sum_{t=1}^{T}[(1-1/e)F_t(x^*) - F_t(y_t)]$$

$$= \sum_{t=1}^{T}[(1-1/e)F_t(x^*) - (1-1/e)F_t(x_\delta^*) + (1-1/e)F_t(x_\delta^*) - F_t(y_t)]$$

$$= (1-1/e)\sum_{t=1}^{T}[F_t(x^*) - F_t(x_\delta^*)] + \sum_{t=1}^{T}[(1-1/e)\hat{F}_t(x_\delta^*) - \hat{F}_t(y_t)]$$

$$\left.\rule{0pt}{0pt}\right]$$

$$+ \sum_{t=1}^{T}(1-1/e)[F_t(x_\delta^*) - \hat{F}_t(x_\delta^*)] - \sum_{t=1}^{T}[F_t(y_t) - \hat{F}_t(y_t)]$$

$$\leq (1-1/e)\sum_{t=1}^{T}[F_t(x^*) - F_t(x_\delta^*)] + \sum_{t=1}^{T}[(1-1/e)\hat{F}_t(x_\delta^*) - \hat{F}_t(y_t)] + T(1-1/e)L_1\delta + TL_1\delta$$

$$= (1-1/e)\sum_{t=1}^{T}[F_t(x^*) - F_t(x_\delta^*)] + \sum_{t=1}^{T}[(1-1/e)\hat{F}_t(x_\delta^*) - \hat{F}_t(y_t)] + (2-1/e)L_1T\delta.$$

Suppose $x' \in \mathcal{K}'$ such that $\|x^* - x'\| = d(x^*, x') = d(x^*, \mathcal{K}') \leq d(\mathcal{K}, \mathcal{K}') \leq c_1\delta^\gamma$, then we have

$$\sum_{t=1}^{T}[F_t(x^*) - F_t(x_\delta^*)] = \sum_{t=1}^{T}[F_t(x^*) - F_t(x') + F_t(x') - F_t(x_\delta^*)]$$

$$= \sum_{t=1}^{T}[F_t(x^*) - F_t(x')] + [\sum_{t=1}^{T}F_t(x') - \sum_{t=1}^{T}F_t(x_\delta^*)]$$

$$\leq \sum_{t=1}^{T}[L_1\|x^* - x'\|] + 0$$

$$\leq c_1L_1T\delta^\gamma,$$

where the first inequality holds thanks to the optimality of $x_\delta^*$ and the assumption that $F_t$ is $L_1$-Lipschitz.

Moreover, we have

$$\hat{\mathcal{R}}_T \triangleq \sum_{t=1}^{T}[(1-1/e)\hat{F}_t(x_\delta^*) - \hat{F}_t(y_t)]$$

$$= \sum_{q=1}^{Q}\sum_{i=1}^{L}[(1-1/e)\hat{F}_{t_{q,i}}(x_\delta^*) - \hat{F}_{t_{q,i}}(x_q)] + \sum_{q=1}^{Q}\sum_{k=1}^{K}[\hat{F}_{t_{q,k}}(x_q) - \hat{F}_{t_{q,k}}(y_{t_{q,k}})]$$

$$\leq \sum_{q=1}^{Q}\sum_{i=1}^{L}[(1-1/e)\hat{F}_{t_{q,i}}(x_\delta^*) - \hat{F}_{t_{q,i}}(x_q)] + \sum_{q=1}^{Q}\sum_{k=1}^{K}[2M_1]$$

$$= \sum_{q=1}^{Q}\sum_{i=1}^{L}[(1-1/e)\hat{F}_{t_{q,i}}(x_\delta^*) - \hat{F}_{t_{q,i}}(x_q)] + 2M_1QK$$

where the inequality holds since

$$|\hat{F}_{q,t_k}(x)| = |\mathbb{E}_{v\sim B^n}[F_{q,t_k}(x+\delta v)]| \leq \mathbb{E}[|F_{q,t_k}(x+\delta v)|] \leq M_1.$$

So by now, we have

$$\mathcal{R}_T \leq (1-1/e)c_1L_1T\delta^\gamma + (2-1/e)L_1T\delta + 2M_1QK + \sum_{q=1}^{Q}\sum_{i=1}^{L}[(1-1/e)\hat{F}_{t_{q,i}}(x_\delta^*) - \hat{F}_{t_{q,i}}(x_q)].$$

In order to upper bound $\sum_{q=1}^{Q}\sum_{i=1}^{L}[(1-1/e)\hat{F}_{t_{q,i}}(x_\delta^*) - \hat{F}_{t_{q,i}}(x_q)]$, we first define the average function:

$$\bar{F}_{q,k}(x) = \frac{\sum_{i=k+1}^{L}\hat{F}_{t_{q,i}}(x)}{L-k}.$$

Recall that $(t_{q,1}, \cdots, t_{q,K})$ is a random sub-sequence of $\{(q-1)L+1, \cdots, qL\}$, and is used for "exploration".

We first claim that similar result to Lemma 3 in Appendix B still holds for Algorithm 2.

**Lemma 9.** *If $F_t$ is monotone continuous DR-submodular and $L_2$-smooth, $x_t^{(k+1)} = x_t^{(k)} + \frac{1}{K}(v_t^{(k)} - \underline{u})$ for $k \in [K]$, where $v_t^{(k)}, x_t^{(k)} \in \mathcal{K}', \underline{u}$ is the lower bound of $\mathcal{K}'$, then*

$$F_t(x_\delta^*) - F_t(x_t^{(k+1)}) \leq (1-1/K)[F_t(x_\delta^*) - F_t(x_t^{(k)})]$$
$$-\frac{1}{K}[-\frac{1}{2\beta^{(k)}}\|\nabla F_t(x_t^{(k)}) - d_t^{(k)}\|^2 - \frac{\beta^{(k)}D^2}{2} + \langle d_t^{(k)}, v_t^{(k)} - x_\delta^*\rangle] + \frac{L_2D^2}{2K^2},$$

*where $\{\beta^{(k)}\}$ is a sequence of positive parameters to be determined.*

*Proof of Lemma 9.* Since $F_t$ is $L_2$-smooth and $x_t^{(k+1)} = x_t^{(k)} + \frac{1}{K}(v_t^{(k)} - \underline{u})$, we have

$$F_t(x_t^{(k+1)}) \geq F_t(x_t^{(k)}) + \langle \nabla F_t(x_t^{(k)}), x_t^{(k+1)} - x_t^{(k)}\rangle - \frac{L_2}{2}\|x_t^{(k+1)} - x_t^{(k)}\|^2$$

$$= F_t(x_t^{(k)}) + \langle \frac{1}{K}\nabla F_t(x_t^{(k)}), v_t^{(k)} - \underline{u}\rangle - \frac{L_2}{2K^2}\|v_t^{(k)} - \underline{u}\|^2 \qquad (19)$$

$$\geq F_t(x_t^{(k)}) + \frac{1}{K}\langle \nabla F_t(x_t^{(k)}), v_t^{(k)} - \underline{u}\rangle - \frac{L_2D^2}{2K^2}.$$

We can rewrite the term $\langle \nabla F_t(x_t^{(k)}), v_t^{(k)} - \underline{u}\rangle$ as

$$\langle \nabla F_t(x_t^{(k)}), v_t^{(k)} - \underline{u}\rangle = \langle \nabla F_t(x_t^{(k)}) - d_t^{(k)}, v_t^{(k)}\rangle + \langle d_t^{(k)}, v_t^{(k)}\rangle - \langle \nabla F_t(x_t^{(k)}), \underline{u}\rangle$$
$$= \langle \nabla F_t(x_t^{(k)}) - d_t^{(k)}, v_t^{(k)} - x_\delta^*\rangle + \langle \nabla F_t(x_t^{(k)}) - d_t^{(k)}, x_\delta^*\rangle$$
$$+ \langle d_t^{(k)}, v_t^{(k)}\rangle - \langle \nabla F_t(x_t^{(k)}), \underline{u}\rangle$$
$$= \langle \nabla F_t(x_t^{(k)}) - d_t^{(k)}, v_t^{(k)} - x_\delta^*\rangle + \langle \nabla F_t(x_t^{(k)}), x_\delta^* - \underline{u}\rangle + \langle d_t^{(k)}, v_t^{(k)} - x_\delta^*\rangle.$$
$$(20)$$

Denote $y_\delta^* = x_\delta^* - \underline{u}, y_t^{(k)} = x_t^{(k)} - \underline{u}$, then $y_\delta^* \geq 0, y_t^{(k)} \geq 0$, by the definition of lower bound $\underline{u}$, and the fact $x_\delta^*, x_t^{(k)} \in \mathcal{K}'$. Since $F_t$ is monotone and is concave along non-negative directions, we have

$$F_t(x_\delta^*) - F_t(x_t^{(k)}) = F_t(y_\delta^* + \underline{u}) - F_t(y_t^{(k)} + \underline{u})$$
$$\leq F_t[(y_\delta^* + \underline{u}) \vee (y_t^{(k)} + \underline{u})] - F_t(y_t^{(k)} + \underline{u})$$
$$\leq \langle \nabla F_t(y_t^{(k)} + \underline{u}), [(y_\delta^* + \underline{u}) \vee (y_t^{(k)} + \underline{u})] - (y_t^{(k)} + \underline{u})\rangle$$
$$= \langle \nabla F_t(y_t^{(k)} + \underline{u}), [(y_\delta^* + \underline{u}) - (y_t^{(k)} + \underline{u})] \vee 0\rangle \qquad (21)$$
$$= \langle \nabla F_t(y_t^{(k)} + \underline{u}), (y_\delta^* - y_t^{(k)}) \vee 0\rangle$$
$$\leq \langle \nabla F_t(y_t^{(k)} + \underline{u}), y_\delta^*\rangle$$
$$= \langle \nabla F_t(x_t^{(k)}), x_\delta^* - \underline{u}\rangle.$$

Combine Eqs. (20) and (21), we have

$$\langle \nabla F_t(x_t^{(k)}), v_t^{(k)} - \underline{u}\rangle \geq \langle \nabla F_t(x_t^{(k)}) - d_t^{(k)}, v_t^{(k)} - x_\delta^*\rangle + [F_t(x_\delta^*) - F_t(x_t^{(k)})] + \langle d_t^{(k)}, v_t^{(k)} - x_\delta^*\rangle. \quad (22)$$

By Young's ineqaulity, we have

$$\langle \nabla F_t(x_t^{(k)}) - d_t^{(k)}, v_t^{(k)} - x_\delta^* \rangle \geq -\frac{1}{2\beta^{(k)}} \| \nabla F_t(x_t^{(k)}) - d_t^{(k)} \|^2 - \frac{\beta^{(k)}}{2} \| v_t^{(k)} - x_\delta^* \|^2$$

$$\geq -\frac{1}{2\beta^{(k)}} \| \nabla F_t(x_t^{(k)}) - d_t^{(k)} \|^2 - \frac{\beta^{(k)} D^2}{2}. \tag{23}$$

Now combine Eqs. (19), (22) and (23), we have

$$F_t(x_t^{(k+1)}) \geq \frac{1}{K} \left[ -\frac{1}{2\beta^{(k)}} \| \nabla F_t(x_t^{(k)}) - d_t^{(k)} \|^2 - \frac{\beta^{(k)} D^2}{2} + [F_t(x_\delta^*) - F_t(x_t^{(k)})] + \langle d_t^{(k)}, v_t^{(k)} - x_\delta^* \rangle \right]$$

$$+ F_t(x_t^{(k)}) - \frac{L_2 D^2}{2K^2}.$$

Or, equivalently,

$$F_t(x_\delta^*) - F_t(x_t^{(k+1)}) \leq (1 - 1/K)[F_t(x_\delta^*) - F_t(x_t^{(k)})]$$

$$- \frac{1}{K} \left[ -\frac{1}{2\beta^{(k)}} \| \nabla F_t(x_t^{(k)}) - d_t^{(k)} \|^2 - \frac{\beta^{(k)} D^2}{2} + \langle d_t^{(k)}, v_t^{(k)} - x_\delta^* \rangle \right] + \frac{L_2 D^2}{2K^2},$$

$$\square$$

Since $\hat{F}_t$ is monotone continuous DR-submodular and $L_2$-smooth for all $t$, with Lemma 9, and repeating the proof of Lemma 4 in Appendix B, we have

$$\mathbb{E}[(1 - 1/e)\bar{F}_{q,0}(x_\delta^*) - \bar{F}_{q,0}(x_q)] \leq \mathbb{E}\left[ \frac{1}{K} \sum_{k=1}^{K} \left[ \frac{1}{2\beta^{(k)}} \Delta_q^{(k)} + \frac{\beta^{(k)} D^2}{2} \right] \right] + \frac{L_2 D^2}{2K}$$

$$+ 1/K \sum_{k=1}^{K} (1 - 1/K)^{K-k} \mathbb{E}[\langle d_q^{(k)}, x_\delta^* - v_q^{(k)} \rangle]$$

where $\Delta_q^{(k)} = \| \nabla \bar{F}_{q,k-1}(x_q^{(k)}) - d_q^{(k)} \|^2$.

Therefore, we have

$$\mathbb{E}\left[ \sum_{q=1}^{Q} \sum_{i=1}^{L} [(1 - 1/e)\hat{F}_{t_{q,i}}(x_\delta^*) - \hat{F}_{t_{q,i}}(x_q)] \right]$$

$$= \sum_{q=1}^{Q} L \mathbb{E}[(1 - 1/e)\bar{F}_{q,0}(x_\delta^*) - \bar{F}_{q,0}(x_q)]$$

$$= \mathbb{E}\left[ \frac{L}{K} \sum_{q=1}^{Q} \sum_{k=1}^{K} \frac{\Delta_q^{(k)}}{2\beta^{(k)}} \right] + \frac{LQ}{K} \sum_{k=1}^{K} \frac{\beta^{(k)} D^2}{2} + \frac{LQL_2 D^2}{2K}$$

$$+ \frac{L}{K} \sum_{k=1}^{K} (1 - 1/K)^{K-k} \sum_{q=1}^{Q} \mathbb{E}[\langle d_q^{(k)}, x_\delta^* - v_q^{(k)} \rangle] \tag{24}$$

$$\leq \mathbb{E}\left[ \frac{L}{K} \sum_{q=1}^{Q} \sum_{k=1}^{K} \frac{\Delta_q^{(k)}}{2\beta^{(k)}} \right] + \frac{LQ}{K} \sum_{k=1}^{K} \frac{\beta^{(k)} D^2}{2} + \frac{LQL_2 D^2}{2K}$$

$$+ \frac{L}{K} \sum_{k=1}^{K} 1 \cdot \mathcal{R}_Q^{\mathcal{E}}$$

$$\leq \mathbb{E}\left[ \frac{L}{K} \sum_{q=1}^{Q} \sum_{k=1}^{K} \frac{\Delta_q^{(k)}}{2\beta^{(k)}} \right] + \frac{LQ}{K} \sum_{k=1}^{K} \frac{\beta^{(k)} D^2}{2} + \frac{LQL_2 D^2}{2K} + L\mathcal{R}_Q^{\mathcal{E}}.$$

Then we have

$$\mathbb{E}[\mathcal{R}_T] \leq (1-1/e)c_1 L_1 T \delta^\gamma + (2-1/e)L_1 T \delta + 2M_1 QK$$

$$+ \mathbb{E}\left[\frac{L}{K}\sum_{q=1}^{Q}\sum_{k=1}^{K}\frac{\Delta_q^{(k)}}{2\beta^{(k)}}\right] + \frac{LQ}{K}\sum_{k=1}^{K}\frac{\beta^{(k)}D^2}{2} + \frac{LQL_2 D^2}{2K} + LR_Q^{\mathcal{E}}. \tag{25}$$

Note $\mathcal{R}_Q^{\mathcal{E}}$ is the regret of the online linear maximization oracle $\mathcal{E}$ at horizon $Q$, which is of order $O(\sqrt{Q})$. So in order to get an upper bound for the expected regret of Algorithm 2, the key is to bound $\mathbb{E}[\Delta_q^{(k)}]$. Here, we have an analogue of Lemma 5 in Appendix B:

**Lemma 10.** *Under the setting of Theorem 4, we have*

$$\mathbb{E}[\Delta_q^{(k)}] \leq \rho_k^2 \sigma^2 + (1-\rho_k)^2 \mathbb{E}[\Delta_q^{(k-1)}] + (1-\rho_k)^2 \frac{G}{(k+2)^2} + (1-\rho_k)^2 \left[\frac{G}{\alpha_k(k+2)^2} + \alpha_k \mathbb{E}[\Delta_q^{(k-1)}]\right],$$

*where $\{\alpha_k\}$ is a sequence of positive parameters to be determined, $\sigma^2 = L_1^2 + \frac{d^2 M_1^2}{\delta^2}$, $G = [3L_2 R + 2L_1]^2$.*

*Proof of Lemma 10.* First, the decomposition of $\Delta_q^{(k)}$ Eq. (7) still holds, with $\tilde{\nabla}F_{t_{q,k}}(x_q^{(k)})$ replaced by $g_{q,k}$.

We also denote $\mathcal{F}_{q,k}$ to be the $\sigma$-field generated by $t_{q,1}, t_{q,2}, \cdots, t_{q,k}$. Since $\mathbb{E}[g_{q,k}|\mathcal{F}_{q,k}] = \nabla\hat{F}_{t_{q,k}}(x_q^{(k)})|\mathcal{F}_{q,k}$, we have $\mathbb{E}[g_{q,k}|\mathcal{F}_{q,k-1}] = \nabla\bar{F}_{q,k-1}(x_q^{(k)})|\mathcal{F}_{q,k-1}$. Then by law of iterated expectations, we can get the results similar to Eqs. (8) to (12).

Precisely, we have:

$$\mathbb{E}[\mathbb{E}[\|\nabla\bar{F}_{q,k-1}(x_q^{(k)}) - \nabla\hat{F}_{t_{q,k}}(x_q^{(k)})\|^2|\mathcal{F}_{q,k-1}]] = \mathbb{E}[\text{Var}(\nabla\hat{F}_{t_{q,k}}(x_q^{(k)})|\mathcal{F}_{q,k-1})]$$
$$\leq \mathbb{E}[\|\nabla\hat{F}_{t_{q,k}}(x_q^{(k)})\|^2]$$
$$\leq L_1^2,$$

$$\mathbb{E}[\mathbb{E}[\|\nabla\hat{F}_{t_{q,k}}(x_q^{(k)}) - g_{q,k}\|^2|\mathcal{F}_{q,k-1}]] = \mathbb{E}[\|\nabla\hat{F}_{t_{q,k}}(x_q^{(k)}) - g_{q,k}\|^2]$$
$$= \mathbb{E}[\mathbb{E}[\|\nabla\hat{F}_{t_{q,k}}(x_q^{(k)}) - g_{q,k}\|^2|\mathcal{F}_{q,k}]]$$
$$= \mathbb{E}[\text{Var}(g_{q,k}|\mathcal{F}_{q,k})]$$
$$\leq \frac{d^2 M_1^2}{\delta^2},$$

and

$$\mathbb{E}[\mathbb{E}[\langle\nabla\bar{F}_{q,k-1}(x_q^{(k)}) - \nabla\hat{F}_{t_{q,k}}(x_q^{(k)}), \nabla\hat{F}_{t_{q,k}}(x_q^{(k)}) - g_{q,k}\rangle|\mathcal{F}_{q,k-1}]] = 0.$$

Thus we have

$$\mathbb{E}[\|\nabla\bar{F}_{q,k-1}(x_q^{(k)}) - g_{q,k}\|^2]$$
$$= \mathbb{E}[\mathbb{E}[\|\nabla\bar{F}_{q,k-1}(x_q^{(k)}) - g_{q,k}\|^2|\mathcal{F}_{q,k-1}]]$$
$$= \mathbb{E}[\mathbb{E}[\|\nabla\bar{F}_{q,k-1}(x_q^{(k)}) - \nabla\hat{F}_{t_{q,k}}(x_q^{(k)})\|^2 + \|\nabla\hat{F}_{t_{q,k}}(x_q^{(k)}) - g_{q,k}\|^2$$
$$+ 2\langle\nabla\bar{F}_{q,k-1}(x_q^{(k)}) - \nabla\hat{F}_{t_{q,k}}(x_q^{(k)}), \nabla\hat{F}_{t_{q,k}}(x_q^{(k)}) - g_{q,k}\rangle|\mathcal{F}_{q,k-1}]] \tag{26}$$
$$\leq L_1^2 + \frac{d^2 M_1^2}{\delta^2}$$
$$\triangleq \sigma^2.$$

We also have the results similar to Eqs. (13) and (14):

$$\mathbb{E}[\langle\nabla\bar{F}_{q,k-1}(x_q^{(k)}) - g_{q,k}, \nabla\bar{F}_{q,k-1}(x_q^{(k)}) - \nabla\bar{F}_{q,k-2}(x_q^{(k-1)})\rangle] = 0, \tag{27}$$

and

$$\mathbb{E}[\langle\nabla\bar{F}_{q,k-1}(x_q^{(k)}) - g_{q,k}, \nabla\bar{F}_{q,k-2}(x_q^{(k-1)}) - d_q^{(k-1)}\rangle] = 0. \tag{28}$$

Also, by Young's Inequality, we have

$$\langle \nabla \bar{F}_{q,k-1}(x_q^{(k)}) - \nabla \bar{F}_{q,k-2}(x_q^{(k-1)}), \nabla \bar{F}_{q,k-2}(x_q^{(k-1)}) - d_q^{(k-1)} \rangle$$
$$\leq \frac{1}{2\alpha_k} \| \nabla \bar{F}_{q,k-1}(x_q^{(k)}) - \nabla \bar{F}_{q,k-2}(x_q^{(k-1)}) \|^2 + \frac{\alpha_k}{2} \Delta_q^{(k-1)}. \tag{29}$$

Now we turn to bound $\| \nabla \bar{F}_{q,k-1}(x_q^{(k)}) - \nabla \bar{F}_{q,k-2}(x_q^{(k-1)}) \|^2 \triangleq z_{q,k}^2$. Actually, we have

$$\mathbb{E}[z_{q,k}^2] = \mathbb{E}[\mathbb{E}[\| \nabla \bar{F}_{q,k-1}(x_q^{(k)}) - \nabla \bar{F}_{q,k-2}(x_q^{(k-1)}) \|^2 | \mathcal{F}_{q,k-2}]]$$

$$= \mathbb{E}[\mathbb{E}[\| \frac{\sum_{i=k}^{L} \nabla \hat{F}_{t_q,i}(x_q^{(k)})}{L-k+1} - \frac{\sum_{i=k-1}^{L} \nabla \hat{F}_{t_q,i}(x_q^{(k-1)})}{L-k+2} \|^2 | \mathcal{F}_{q,k-2}]]$$

$$= \mathbb{E}[\mathbb{E}[\| \frac{\sum_{i=k}^{L} \nabla \hat{F}_{t_q,i}(x_q^{(k)}) - \nabla \hat{F}_{t_q,i}(x_q^{(k-1)})}{L-k+2} + \frac{\sum_{i=k}^{L} \nabla \hat{F}_{t_q,i}(x_q^{(k)})}{(L-k+1)(L-k+2)}$$

$$- \frac{\nabla \hat{F}_{t_{q,k-1}}(x_q^{(k-1)})}{L-k+2} \|^2 | \mathcal{F}_{q,k-2}]]$$

$$\leq \mathbb{E}[\mathbb{E}[(\sum_{i=k}^{L} \| \frac{\nabla \hat{F}_{t_q,i}(x_q^{(k)}) - \nabla \hat{F}_{t_q,i}(x_q^{(k-1)})}{L-k+2} \| + \sum_{i=k}^{L} \| \frac{\nabla \hat{F}_{t_q,i}(x_q^{(k)})}{(L-k+1)(L-k+2)} \|$$

$$+ \| \frac{\nabla \hat{F}_{t_{q,k-1}}(x_q^{(k-1)})}{L-k+2} \|)^2 | \mathcal{F}_{q,k-2}]],$$

where the inequality comes from the Triangle Inequality of norms.

Recall the update rule where $x_q^{(k)} = x_q^{(k-1)} + \frac{1}{K}(v_q^{(k-1)} - \underline{u})$ and that $\hat{F}_t$ is $L_2$-smooth, we have

$$\| \nabla \hat{F}_{t_q,i}(x_q^{(k)}) - \nabla \hat{F}_{t_q,i}(x_q^{(k-1)}) \| \leq L_2 \frac{\| v_q^{(k-1)} - \underline{u} \|}{K} \leq \frac{L_2 R}{K}.$$

Also by Assumption 2, $\| \nabla F_{t_q,i}(x) \| \leq L_1$ for all $x \in \mathcal{K}$, thus $\| \nabla \hat{F}_{t_q,i}(x_q^{(k)}) \| \leq L_1$, $\| \nabla \hat{F}_{t_{q,k-1}}(x_q^{(k-1)}) \| \leq L_1$. Therefore, we have

$$\mathbb{E}[z_{q,k}^2] \leq [(L-k+1)\frac{L_2 R}{K} \frac{1}{L-k+2} + (L-k+1)\frac{L_1}{(L-k+1)(L-k+2)} + \frac{L_1}{L-k+2}]^2$$

$$\leq \left( \frac{L-k+1}{L-k+2} \frac{L_2 R}{K} + \frac{2L_1}{L-k+2} \right)^2.$$

Since we assume $L \gg K$, we can always choose $L, K$ such that $L \geq 2K$. So we have $\frac{2L_1}{L-k+2} \leq \frac{2L_1}{2K-k+2} \leq \frac{2L_1}{K+2} \leq \frac{2L_1}{k+2}$. Also, $\frac{L-k+1}{L-k+2} \frac{L_2 R}{K} \leq \frac{L_2 R}{K} = \frac{K+2}{K} \frac{L_2 R}{K+2} \leq 3\frac{L_2 R}{K+2} \leq \frac{3L_2 R}{k+2}$.

Therefore, we have

$$\mathbb{E}[z_{q,k}^2] \leq \left( \frac{3L_2 R}{k+2} + \frac{2L_1}{k+2} \right)^2$$

$$= \left( \frac{3L_2 R + 2L_1}{k+2} \right)^2 \tag{30}$$

$$\triangleq \frac{G}{(k+2)^2}.$$

Combining Eqs. (26) to (30), we have

$$\mathbb{E}[\Delta_q^{(k)}] \leq \rho_k^2 \sigma^2 + (1-\rho_k)^2 \mathbb{E}[\Delta_q^{(k-1)}] + (1-\rho_k)^2 \frac{G}{(k+2)^2} + (1-\rho_k)^2 \left[ \frac{G}{\alpha_k(k+2)^2} + \alpha_k \mathbb{E}[\Delta_q^{(k-1)}] \right].$$

$$\square$$

Applying Lemma 10 and setting $\alpha_k = \frac{\rho_k}{2}, \forall k \in 1, 2, \cdots, K$, we have

$$\mathbb{E}[\Delta_q^{(k)}] \leq \rho_k^2 \sigma^2 + (1 - \rho_k)^2 \mathbb{E}[\Delta_q^{(k-1)}] + (1 - \rho_k)^2 \frac{G}{(k+2)^2}$$

$$+ (1 - \rho_k)^2 \left[ \frac{G}{\alpha_k (k+2)^2} + \alpha_k \mathbb{E}[\Delta_q^{(k-1)}] \right]$$

$$= \rho_k^2 \sigma^2 + \frac{G}{(k+2)^2}(1 - \rho_k)^2 \left( 1 + \frac{2}{\rho_k} \right) + \mathbb{E}[\Delta_q^{(k-1)}](1 - \rho_k)^2 \left( 1 + \frac{\rho_k}{2} \right).$$

Note that if $0 < \rho_k \leq 1$, then we have

$$(1 - \rho_k)^2 \left( 1 + \frac{2}{\rho_k} \right) \leq \left( 1 + \frac{2}{\rho_k} \right)$$

and

$$(1 - \rho_k)^2 \left( 1 + \frac{\rho_k}{2} \right) \leq (1 - \rho_k).$$

So in this case, we have

$$\mathbb{E}[\Delta_q^{(k)}] \leq \rho_k^2 \sigma^2 + \frac{G}{(k+2)^2} \left( 1 + \frac{2}{\rho_k} \right) + \mathbb{E}[\Delta_q^{(k-1)}](1 - \rho_k). \tag{31}$$

**Lemma 11.** *Under the setting of Theorem 4, we have*

$$\mathbb{E}[\Delta_q^{(k)}] \leq \frac{N_0}{(k+3)^{2/3}}, \forall k \in [K],$$

*where $N_0 = 4^{2/3}(2\sigma^2 + G)$.*

*Proof of Lemma 11.* Since $\rho_k = \frac{2}{(k+2)^{2/3}}$, we have $0 < \rho_k \leq 1$, and

$$\mathbb{E}[\Delta_q^{(k)}] \leq \frac{4\sigma^2}{(k+2)^{4/3}} + \frac{G}{(k+2)^2}[1 + (k+2)^{2/3}] + \mathbb{E}[\Delta_q^{(k-1)}] \left( 1 - \frac{2}{(k+2)^{2/3}} \right)$$

$$\leq \frac{4\sigma^2}{(k+2)^{4/3}} + \frac{G}{(k+2)^{4/3}} + \frac{G}{(k+2)^{4/3}} + \mathbb{E}[\Delta_q^{(k-1)}] \left( 1 - \frac{2}{(k+2)^{2/3}} \right)$$

$$= \frac{4\sigma^2 + 2G}{(k+2)^{4/3}} + \mathbb{E}[\Delta_q^{(k-1)}] \left( 1 - \frac{2}{(k+2)^{2/3}} \right)$$

$$\leq \frac{\frac{4^{2/3}}{2}(4\sigma^2 + 2G)}{(k+2)^{4/3}} + \mathbb{E}[\Delta_q^{(k-1)}] \left( 1 - \frac{2}{(k+2)^{2/3}} \right)$$

$$= \frac{4^{2/3}(2\sigma^2 + G)}{(k+2)^{4/3}} + \mathbb{E}[\Delta_q^{(k-1)}] \left( 1 - \frac{2}{(k+2)^{2/3}} \right)$$

$$\triangleq \frac{N_0}{(k+2)^{4/3}} + \mathbb{E}[\Delta_q^{(k-1)}] \left( 1 - \frac{2}{(k+2)^{2/3}} \right).$$

Recall that $\Delta_q^{(k)} = \|\nabla \bar{F}_{q,k-1}(x_q^{(k)}) - d_q^{(k)}\|^2$, and thus

$$\Delta_q^{(1)} = \|\nabla \bar{F}_{q,0}(\underline{u}) - d_q^{(1)}\|^2$$

$$= \|\frac{\sum_{i=1}^L \nabla \hat{F}_{t_{q,i}}(\underline{u})}{L} - \frac{2}{3^{2/3}}g_{q,1})\|^2$$

$$\leq \left( \sum_{i=1}^L \|\frac{\nabla \hat{F}_{t_{q,i}}(\underline{u})}{L}\| + \|\frac{2}{3^{2/3}}g_{q,1}\| \right)^2$$

$$\leq \left( L\frac{L_1}{L} + \frac{2}{3^{2/3}}\frac{d}{\delta}M_1 \right)^2$$

$$\leq (L_1 + \frac{d}{\delta}M_1)^2.$$

Now we claim that $\mathbb{E}[\Delta_q^{(k)}] \leq \frac{N_0}{(k+3)^{2/3}}$ for any $k \in [K]$. We prove it by induction. When $k = 1$, we have

$$\frac{N_0}{(1+3)^{2/3}} = 2\sigma^2 + G \geq 2\sigma^2 = 2(L_1^2 + \frac{d^2 M_1^2}{\delta^2}) \geq (L_1 + \frac{dM_1}{\delta})^2 \geq \Delta_q^{(1)},$$

where the second inequality holds since $2(a^2 + b^2) \geq (a+b)^2$.

Assume the statement holds for $k-1$, i.e., $\mathbb{E}[\Delta_q^{(k-1)}] \leq \frac{N_0}{(k+2)^{2/3}}$, then

$$\begin{aligned}
\mathbb{E}[\Delta_q^{(k)}] &\leq \frac{N_0}{(k+2)^{4/3}} + \mathbb{E}[\Delta_q^{(k-1)}]\left(1 - \frac{2}{(k+2)^{2/3}}\right) \\
&\leq \frac{N_0}{(k+2)^{4/3}} + \frac{N_0}{(k+2)^{2/3}}\left(1 - \frac{2}{(k+2)^{2/3}}\right) \\
&= \frac{N_0[(k+2)^{2/3} - 1]}{(k+2)^{4/3}}.
\end{aligned}$$

Since $(k+3)^2 = k^2 + 6k + 9 \leq k^2 + 4k + 4 + 1 + 3(k+2) \leq (k+2)^2 + 1 + 3(k+2)^{4/3} + 3(k+2)^{2/3} = [(k+2)^{2/3} + 1]^3$, by taking the cube roots of both sides, we have $(k+3)^{2/3} \leq (k+2)^{2/3} + 1$, which implies that $[(k+2)^{2/3} - 1](k+3)^{2/3} \leq [(k+2)^{2/3} - 1][(k+2)^{2/3} + 1] \leq (k+2)^{4/3}$, i.e., $\frac{(k+2)^{2/3}-1}{(k+2)^{4/3}} \leq \frac{1}{(k+3)^{2/3}}$. Thus we have

$$\mathbb{E}[\Delta_q^{(k)}] \leq \frac{N_0}{(k+3)^{2/3}}, \forall k \in [K].$$

$\square$

Recall that in Eq. (25), we have

$$\begin{aligned}
\mathbb{E}[\mathcal{R}_T] &\leq (1 - 1/e)c_1 L_1 T\delta^\gamma + (2 - 1/e)L_1 T\delta + 2M_1 QK \\
&+ \mathbb{E}[\frac{L}{K}\sum_{q=1}^{Q}\sum_{k=1}^{K}\frac{\Delta_q^{(k)}}{2\beta^{(k)}}] + \frac{LQ}{K}\sum_{k=1}^{K}\frac{\beta^{(k)}D^2}{2} + \frac{LQL_2 D^2}{2K} + L\mathcal{R}_Q^{\mathcal{E}}.
\end{aligned}$$

So if we set $\beta^{(k)} = \frac{1}{\delta(k+3)^{1/3}}$, then by Lemma 11, we have

$$\sum_{k=1}^{K}\frac{\mathbb{E}[\Delta_q^{(k)}]}{\beta^{(k)}} \leq \sum_{k=1}^{K}\frac{\delta N_0}{(k+3)^{1/3}} \leq \sum_{k=1}^{K}\frac{\delta N_0}{k^{1/3}} \leq \int_0^K \frac{\delta N_0}{x^{1/3}}\mathrm{d}x = \frac{3\delta N_0}{2}K^{2/3}.$$

Similarly,

$$\sum_{k=1}^{K}\beta^{(k)} = \sum_{k=1}^{K}\frac{1}{\delta(k+3)^{1/3}} \leq \frac{3K^{2/3}}{2\delta}.$$

Therefore, we have

$$\mathbb{E}[\mathcal{R}_T] \leq (1-1/e)c_1 L_1 T\delta^\gamma + (2-1/e)L_1 T\delta + 2M_1 QK + \frac{3\delta N_0 LQ}{4K^{1/3}} + \frac{3D^2 LQ}{4\delta K^{1/3}} + \frac{LQL_2 D^2}{2K} + L\mathcal{R}_Q^{\mathcal{E}}.$$

By setting $\delta = c_2 T^{-\frac{1}{3+6\min\{1,\gamma\}}}, Q = T^{\frac{2\min\{1,\gamma\}}{3+6\min\{1,\gamma\}}}, L = T^{\frac{3+4\min\{1,\gamma\}}{3+6\min\{1,\gamma\}}}, K = T^{\frac{1+\min\{1,\gamma\}}{1+2\min\{1,\gamma\}}}$, and recall that $\mathcal{R}_Q^{\mathcal{E}} \leq C\sqrt{Q} = CT^{\frac{\min\{1,\gamma\}}{3+6\min\{1,\gamma\}}}, N_0 = 4^{2/3}(2\sigma^2 + G) = 4^{2/3}(2L_1^2 + \frac{2d^2 M_1^2}{\delta^2} + G)$, where

$G = (3L_2R + 2L_1)^2$ is a constant, we have

$$\mathbb{E}[\mathcal{R}_T] \leq (1 - 1/e)c_1 c_2^\gamma L_1 T^{1 - \frac{\gamma}{3+6\min\{1,\gamma\}}} + (2 - 1/e)c_2 L_1 T^{1 - \frac{1}{3+6\min\{1,\gamma\}}} + 2M_1 T^{\frac{3+5\min\{1,\gamma\}}{3+6\min\{1,\gamma\}}}$$

$$+ \frac{3 \cdot 4^{2/3} c_2 (2L_1^2 + G)}{4} T^{\frac{1+5\min\{1,\gamma\}}{3+6\min\{1,\gamma\}}} + \frac{3 \cdot 4^{2/3} d^2 M_1^2}{2c_2} T^{\frac{3+5\min\{1,\gamma\}}{3+6\min\{1,\gamma\}}} + \frac{3D^2}{4c_2} T^{\frac{3+5\min\{1,\gamma\}}{3+6\min\{1,\gamma\}}}$$

$$+ \frac{L_2 D^2}{2} T^{\frac{\min\{1,\gamma\}}{1+2\min\{1,\gamma\}}} + CT^{\frac{3+5\min\{1,\gamma\}}{3+6\min\{1,\gamma\}}}$$

$$\leq (1 - 1/e)c_1 c_2^\gamma L_1 T^{1 - \frac{\min\{1,\gamma\}}{3+6\min\{1,\gamma\}}} + (2 - 1/e)c_2 L_1 T^{1 - \frac{\min\{1,\gamma\}}{3+6\min\{1,\gamma\}}}$$

$$+ \left[ 2M_1 + \frac{3 \cdot 4^{2/3} d^2 M_1^2}{2c_2} + \frac{3D^2}{4c_2} + C \right] T^{\frac{3+5\min\{1,\gamma\}}{3+6\min\{1,\gamma\}}}$$

$$+ \frac{3 \cdot 4^{2/3} c_2 (2L_1^2 + G)}{4} T^{\frac{1+5\min\{1,\gamma\}}{3+6\min\{1,\gamma\}}} + \frac{L_2 D^2}{2} T^{\frac{\min\{1,\gamma\}}{1+2\min\{1,\gamma\}}}$$

$$= \left[ (1 - 1/e)c_1 c_2^\gamma L_1 + (2 - 1/e)c_2 L_1 + 2M_1 + \frac{3 \cdot 4^{1/6} d^2 M_1^2}{c_2} + \frac{3D^2}{4c_2} + C \right] T^{\frac{3+5\min\{1,\gamma\}}{3+6\min\{1,\gamma\}}}$$

$$+ \frac{3c_2 [2L_1^2 + (3L_2R + 2L_1)^2]}{4^{1/3}} T^{\frac{1+5\min\{1,\gamma\}}{3+6\min\{1,\gamma\}}} + \frac{L_2 D^2}{2} T^{\frac{\min\{1,\gamma\}}{1+2\min\{1,\gamma\}}}.$$

$\square$

## E.2 Down-closed Constraint Set

*Proof of Theorem 2.* Since $\mathcal{K}$ satisfies Assumption 1 and is down-closed, $\alpha = \frac{(\sqrt{d}+1)\delta}{r} = \frac{\sqrt{d}+1}{\sqrt{d}+2} T^{-1/9} < 1$, by Lemma 1, we have Assumption 8 holds with $c_1 = \sqrt{d}(\frac{R}{r} + 1) + \frac{R}{r}, \gamma = 1, \underline{u} = \delta \mathbf{1}$. Then by applying Theorem 4 directly, we can prove Theorem 2. $\square$

## F  Proof of Lemma 2

*Proof of Lemma 2.* We give an example of the matroids which satisfy Lemma 2. Let $\Omega = \{1, 2\}$, the matroid $\mathcal{I} = \{\varnothing, \{1\}, \{2\}\}$. Define set function

$$f(X) = \begin{cases} 0, & X = \varnothing; \\ a, & X = \{1\}; \\ b, & X = \{2\}, \text{ or } X = \{1, 2\}; \end{cases}$$

where $b > a > 0$. It can be verified that $f$ is submodular and its multilinear extension $F(x) = ax_1 + bx_2 - ax_1x_2$, where $x = (x_1, x_2) \in [0, 1]^2$.

Suppose that

$$\text{round}(x) = \begin{cases} \{1\}, & \text{with probability } p_1(x); \\ \{2\}, & \text{with probability } p_2(x); \\ \varnothing, & \text{with probability } p_3(x). \end{cases}$$

Then the assumption $F(x) = \mathbb{E}[f(\text{round}(x)]$ implies $F(x) = p_1(x) \cdot a + p_2(x) \cdot b, \forall b > a > 0$. So we have $p_1(x) = x_1 - x_1x_2, p_2(x) = x_2$.

However, if we define $f$ in another way:

$$f(X) = \begin{cases} 0, & X = \varnothing; \\ b, & X = \{2\}; \\ a, & X = \{1\}, \text{ or } X = \{1, 2\}; \end{cases}$$

where $a > b > 0$. Then it can be also verified that $f$ is submodular and its multilinear extension $F(x) = ax_1 + bx_2 - bx_1x_2$, where $x = (x_1, x_2) \in [0, 1]^2$.

Again, suppose that

$$\text{round}(x) = \begin{cases} \{1\}, & \text{with probability } p_1(x); \\ \{2\}, & \text{with probability } p_2(x); \\ \varnothing, & \text{with probability } p_3(x). \end{cases}$$

Then the assumption $F(x) = \mathbb{E}[f(\text{round}(x)]$ implies $F(x) = p_1(x) \cdot a + p_2(x) \cdot b, \forall a > b > 0$. So we have $p_1(x) = x_1, p_2(x) = x_2 - x_1 x_2$.

Therefore, for different functions $f$'s, we have different sampling schemes $\text{round}(\cdot)$'s, which are subject to the matroid $\mathcal{I}$ constraint, and satisfy $F(x) = \mathbb{E}[f(\text{round}(x)]$, *i.e.*, the sampling scheme does depend on the function. So there does not exist a sampling scheme $\text{round} : [0,1]^d \to \mathcal{I}$, which satisfies $\mathbb{E}[f(\text{round}(x))] = F(x), \forall x \in [0,1]^d$, and does not depend on the submodular set function $f$, $\qquad\qquad\qquad\qquad\qquad\qquad\qquad\qquad\qquad\qquad\qquad\qquad\qquad\qquad\qquad\qquad\square$

## G Proof of Theorem 3

Since Algorithm 3 applies Algorithm 2 on the multilinear extension $F_t$ of $f_t$, a prerequisite is that Assumptions 1, 2 and 4 to 6 all hold. The constraint set $\mathcal{K}$ is a polytope in $[0,1]^d$ that is convex and compact and contains 0. So Assumption 1 holds. Additionally, we have the diameter $D = \sup_{x,y \in \mathcal{K}} \|x - y\| \leq \sqrt{d}$ and the radius $R = \sup_{x \in \mathcal{K}} \|x\| \leq \sqrt{d}$.

Since each objective function $f_t$ is monotone submodular, its multilinear extension $F_t$ is monotone and continuous DR-submodular [16]. If $\sup_{X \subseteq \Omega} |f_t(X)| \leq M$, then Assumption 6 holds for $F_t$ automatically, and the following lemma shows that its multilinear extension $F_t$ is Lipschitz and smooth, which entails Assumption 2.

**Lemma 12** (Lemma 4 in [19]). *For a submodular set function $f$ with $\sup_{X \subseteq \Omega} |f(X)| \leq M$, its multilinear extension $F$ is $(2M\sqrt{d})$-Lipschitz and $(4M\sqrt{d(d-1)})$-smooth.*

In summary, we only need Assumptions 4, 5 and 7. Now we turn to prove Theorem 3.

*Proof of Theorem 3.* We first define $X^* = \arg\max_{X \in \mathcal{I}} \sum_{t=1}^{T} f_t(X)$, the corresponding fractional solution is $\tilde{x} \in \mathcal{K}$, *i.e.*,
$$f_t(X^*) = F_t(\tilde{x}), \tag{32}$$
where $F_t$ is the multilinear extension of $f_t$. We also define $x^* = \arg\max_{x \in \mathcal{K}} \sum_{t=1}^{T} F_t(x)$, $x_\delta^* = \arg\max_{x \in \mathcal{K}'} \sum_{t=1}^{T} F_t(x)$. The $(1 - 1/e)$-regret with horizon $T$ is
$$\mathcal{R}_T = \sum_{t=1}^{T} [(1 - 1/e) f_t(X^*) - f_t(Y_t) \mathbb{1}_{Y_t \in \mathcal{I}}]. \tag{33}$$

We have
$$\sum_{t=1}^{T} f_t(Y_t) \mathbb{1}_{Y_t \in \mathcal{I}} = \sum_{q=1}^{Q} \sum_{i=1}^{L} f_{t_{q,i}}(Y_{t_{q,i}}) \mathbb{1}_{Y_{t_{q,i}} \in \mathcal{I}}$$

$$= \sum_{q=1}^{Q} \sum_{i=K+1}^{L} f_{t_{q,i}}(Y_{t_{q,i}}) + \sum_{q=1}^{Q} \sum_{k=1}^{K} F_{t_{q,k}}(y_{t_{q,k}}) - \sum_{q=1}^{Q} \sum_{k=1}^{K} F_{t_{q,k}}(y_{t_{q,k}})$$

$$+ \sum_{q=1}^{Q} \sum_{k=1}^{K} f_{t_{q,k}}(Y_{t_{q,k}}) \mathbb{1}_{Y_{t_{q,k}} \in \mathcal{I}} \tag{34}$$

$$\geq \sum_{q=1}^{Q} \sum_{i=K+1}^{L} F_{t_{q,i}}(y_{t_{q,i}}) + \sum_{q=1}^{Q} \sum_{k=1}^{K} F_{t_{q,k}}(y_{t_{q,k}}) - \sum_{q=1}^{Q} \sum_{k=1}^{K} M_1 + \sum_{q=1}^{Q} \sum_{k=1}^{K} 0$$

$$= \sum_{t=1}^{T} F_t(y_t) - QKM_1,$$
where the second equation holds since for $t \in \{(q-1)L + 1, \cdots, qL\} \setminus \{t_{q,1}, \cdots, t_{q,K}\}$, $Y_t = \text{LosslessRound}(x_q) \in \mathcal{I}$, and the inequality holds because of the fact that the rounding is lossless and Assumption 7.

Therefore, by Eqs. (32) to (34) and the optimality of $x^*$, we have
$$\mathcal{R}_T \leq \sum_{t=1}^{T} [(1 - 1/e) F_t(\tilde{x}) - F_t(y_t)] + QKM_1 \leq \sum_{t=1}^{T} [(1 - 1/e) F_t(x^*) - F_t(y_t)] + QKM_1. \tag{35}$$

Now we can repeat the proof of Theorem 4 (Appendix E.1) to upper bound $\sum_{t=1}^{T}[(1-1/e)F_t(x^*) - \sum_{t=1}^{T} F_t(y_t)$, with $L_1 = 2M_1\sqrt{d}, L_2 = 4M_1\sqrt{d(d-1)}$ by Lemma 12. The only difference is when we turn to bound $\mathbb{E}[\Delta_q^{(k)}] = \mathbb{E}[\|\nabla\bar{F}_{q,k-1}(x_q^{(k)}) - d_q^{(k)}\|^2]$, where $\bar{F}_{q,k}(x) = \frac{\sum_{i=k+1}^{L}\hat{F}_{t_{q,i}}(x)}{L-k}$, we have a larger upper bound for $\mathbb{E}[\|\nabla\bar{F}_{q,k-1}(x_q^{(k)}) - g_{q,k}\|^2]$, where $g_{q,k} = \frac{d}{\delta}f_{t_{q,k}}(Y_{t_{q,k}})u_{q,k}$. Precisely, we have

$$\mathbb{E}[\|\nabla\bar{F}_{q,k-1}(x_q^{(k)}) - g_{q,k}\|^2]$$

$$=\mathbb{E}[\|\nabla\bar{F}_{q,k-1}(x_q^{(k)}) - \nabla\hat{F}_{t_{q,k}}(x_q^{(k)}) + \nabla\hat{F}_{t_{q,k}}(x_q^{(k)}) - \frac{d}{\delta}F_{t_{q,k}}(y_{t_{q,k}})u_{q,k} + \frac{d}{\delta}F_{t_{q,k}}(y_{t_{q,k}})u_{q,k} - g_{q,k}\|^2]$$

$$=\mathbb{E}[\mathbb{E}[\|\nabla\bar{F}_{q,k-1}(x_q^{(k)}) - \nabla\hat{F}_{t_{q,k}}(x_q^{(k)})\|^2 + \|\nabla\hat{F}_{t_{q,k}}(x_q^{(k)}) - \frac{d}{\delta}F_{t_{q,k}}(y_{t_{q,k}})u_{q,k}\|^2$$

$$+ \|\frac{d}{\delta}F_{t_{q,k}}(y_{t_{q,k}})u_{q,k} - g_{q,k}\|^2$$

$$\leq L_1^2 + \frac{d^2M_1^2}{\delta^2} + \frac{d^2M_1^2}{\delta^2}$$

$$=L_1^2 + \frac{2d^2M_1^2}{\delta^2}$$

$$\triangleq \sigma^2.$$

Plug in the new upper bound for $\sigma^2$, and repeat the analysis of Theorem 4, we have

$$\mathbb{E}[\sum_{t=1}^{T}[(1-1/e)F_t(x^*) - F_t(y_t)]] \leq NT^{\frac{8}{9}} + \frac{3r[2L_1^2 + (3L_2R + 2L_1)^2]}{4^{1/3}(\sqrt{d}+2)}T^{\frac{2}{3}} + \frac{L_2D^2}{2}T^{\frac{1}{3}}, \quad (36)$$

where $N = \frac{(1-1/e)r}{\sqrt{d}+2}[\sqrt{d}(\frac{R}{r}+1) + \frac{R}{r}]L_1 + \frac{(2-1/e)r}{\sqrt{d}+2}L_1 + 2M_1 + \frac{3\cdot4^{2/3}(\sqrt{d}+2)d^2M_1^2}{r} + \frac{3(\sqrt{d}+2)D^2}{4r} + C$, $C$ is a constant satisfying $\mathcal{R}_Q^\mathcal{E} \leq C\sqrt{Q}$.

Combine Eqs. (35) and (36), and using $QKM_1 = M_1T^{8/9}, D \leq \sqrt{d}, R \leq \sqrt{d}$, we conclude

$$\mathbb{E}[\mathcal{R}_T] \leq NT^{\frac{8}{9}} + \frac{3r[2L_1^2 + (3\sqrt{d}L_2 + 2L_1)^2]}{4^{1/3}(\sqrt{d}+2)}T^{\frac{2}{3}} + \frac{L_2d}{2}T^{\frac{1}{3}},$$

where $N = \frac{(1-1/e)r}{\sqrt{d}+2}[\frac{d}{r} + \sqrt{d}(1+\frac{1}{r})]L_1 + \frac{(2-1/e)r}{\sqrt{d}+2}L_1 + 3M_1 + \frac{3\cdot4^{2/3}(\sqrt{d}+2)d^2M_1^2}{r} + \frac{3(\sqrt{d}+2)d}{4r} + C$, $C$ is a constant satisfying $\mathcal{R}_Q^\mathcal{E} \leq C\sqrt{Q}$. $\qquad\square$