[Reviews · NeurIPS 2019]

Reviewer 1



This paper considers the problem of online continuous DR-submodular maximization under full information and bandit feedback settings and also extends the bandit algorithm to the online discrete submodular maximization problem and provides sublinear regret bounds for each case. Originality: This work can be characterized as a novel combination of well-known techniques. This paper builds upon the previous work on online continuous submodular maximization and using the well-known sphere sampling estimator, they provide a one-point unbiased estimation of the gradient for the bandit setting which is the main contribution of this work. All the related work and techniques used in this paper are adequately cited. Quality: This paper provides theoretical analysis for all their claimed results and after doing a high-level check of the proofs, the analysis is technically sound. However, all the proofs are moved to the supplementary file and the main paper is focused on providing a high-level description of the algorithms. I think mentioning the main proof techniques and proof sketches in the main paper could be really beneficial. The paper doesn't provide any experimental results to verify the effectiveness of their proposed algorithms in practice. However, the theoretical contributions of this paper are significant enough to make up for the lack of experiments. Clarity: The paper is extremely well-written and is easy to follow even for one who is not familiar with the prior work on this topic. Significance: As it was mentioned in my response to the previous question, this paper obtains the first sublinear regret bound with a single gradient evaluation per step for the full information setting and additionally, it provides the first sublinear regret bound for the problem in the bandit setting. The main contribution of this paper is the technique used for one-point unbiased gradient estimation in the bandit setting and it is indeed a significant contribution. However, it seems that using other gradient estimation techniques rather than the sphere sampling estimator may lead to better regret bounds (which is an interesting future research direction).

Reviewer 2



-Originality This paper clearly has originality to some extent as the proposed algorithm for OSCM has a different merit from existing ones. However, it seems for me that the methodologies are not new but the authors carefully combine them to construct algorithms. It would be helpful if you push new idea or technique more clearly. -Quality Although I have not verified all the proofs, this paper seems technically correct. At least, the results are reasonable. Please modify some unfinished expressions (e.g., l713 in the supplementary material). It also should be stated that all points the algorithm plays (e.g., y_t in Algorithm 1) belong to the constraint set K. The step sizes \eta_k are parameters in algorithm description, but they should be 1/K (or such that their sum is equal to 1) after all so that the algorithm's choice belong to the constraint set. So the current description may be confusing. -Clarity This paper is well written overall. The authors explain how they construct algorithms step by step. I have the following minor comments. -- It would be helpful for readers if you define some terminologies such as L_1-Lipschitz and matroids. -- Please indicate references for Meta-FW and VR-FW in Table 1. -- Does the definition of radius implicitly use the assumption that the constraint set contains 0? -- l106: it is helpful to clarify the task of linear maximization oracles. -- Algorithm 1: step sizes should be depend on k. -- Lemma 1: the condition "K is down-closed" may be contained in Assumption 5. -- l219: this sentence should be put before l217. -- Theorem 2: Assumption 5 is also used here. -- Algorithm 2&3: how can we know r and delta? Do you assume that it is given? -Significance The problems and results in this paper are of theoretical importance. In particular, the idea of reducing the number of gradient evaluations per function is unique. The main negative is that some motivations appear not strongly convincing. For example, what does Assumption 5 mean in application? In addition, I think RBSM (with a matroid constraint) become more acceptable for wider NeurIPS audience if you mention some application. After the rebuttal: Thank you for your feedback. The application of RBSM seems nice so I raise my score. It will be more convincing with any references for the responsive model (if any).

Reviewer 3



The paper provides interesting techniques for reducing the number of gradient oracle calls in a Frank-Wolfe algorithm. Their main idea (in full-info) is dividing rounds into several subrounds and treating functions in each subround as one virtual function. Although this is a fairly known technique in online convex optimization (OCO), carrying it out in DR-submodular optimization seems nontrivial. Also, estimating a gradient in bandit feedback via querying on sphere is already known in OCO, but combining it with Frank-Wolfe requires some additional efforts. The weaknesses of the paper are: - The resulting regret bounds are worse than the previous O(√T) bound. This is due to subdividing rounds and shows a limitation of their techniques. - Analysis is considerably longer and messier compared to known online DR-submodular maximization algorithms [17, 18]. - Paragraphs are not well structured especially in Section 3: Technical descriptions go on and on without a high level picture. Even though the present paper has several weaknesses, I believe that its technical contributions are strong enough to accept in NeurIPS: especially the first no-regret algorithm in bandit DR-submodular maximization is worth to be published. ----------- update after author response ------------- The authors resolved some concerns raised by the other reviewers. I like the paper and vote for accept.

[Author Response · NeurIPS 2019]

We thank the reviewers for the detailed comments. We will address all the minor issues and do not discuss them individually here. We will also add high-level pictures and proof sketches in the revision.

**Novelty and Originality:** We would like to first describe briefly the key differences between our work and the existing methods.

**(1)** [17, 18] are the only prior work on `OCSM` where up to $K = T^{3/2}$ gradients are required per iteration. Note that $K$ is not a constant, but a function of the total number of rounds $T$. Thus reducing $K$ to 1 is an important step. To do so, we proposed *a series of novel methods* including *the blocking procedure* and *the permutation methods* (L114-131). Besides these, as noted in L137-144, *a novel error analysis* was performed although we relied on the same averaging technique proposed as in [37, 38].
**(2)** Th extension from online setting to bandit is far from trivial given the one-point estimator. Indeed, the bandit information setting is far more challenging than the online (full information) setting and several novel steps are required to design a low-regret algorithm: First, in the bandit algorithm, the point at which we play and the point at which we get the gradient estimation are different (L206-209). To circumvent this issue, we proposed *the biphasic (exploration/exploitation) method* (L210-218). Second, the point for estimation may fall out of the constraint set (L178-179). In [23], this issue was resolved by assuming that $rB^d \subset \mathcal{K} \subset RB^d$, which does not hold for many DR-continuous submodular function, whose domain is defined to be a subset of the non-negative orthant. Therefore we introduced the definition of $\delta$-*interior*, and explained how it can help us address the bandit problem (L180-188). We also proposed a method to *construct proper $\delta$-interior* (L195-197) and prove the result by a geometric analysis (Lemma 1). We established the regret bound based on Lemma 1 (Theorem 2), but also provided the *result for general constraint* (Theorem 4, Lines 219-221).
**(3)** By providing the hardness result (Lemma 2), we showed that it is difficult to extend our method directly from continuous settings to discrete case. Then we considered the `RBSM` model, and established a sublinear regret bound.

**Response to Reviewer #1:**
**Q1:** Using other gradient estimation techniques may lead to better regret bounds. **A1:** We agree with the reviewer and are grateful for suggesting an interesting future research direction.

**Q2:** Provide results for the case that the horizon $T$ is not available offline. **A2:** Using the doubling trick (Auer et al., 1995) will easily extend our methods to the cases where $T$ is unknown. The exploration and exploitation phases are similar to those in Alg. 2, 3.

**Q3:** Provide numerical experiments and compare its running time with previous algorithms for this problem. **A3:** We thank the reviewer for the suggestion. However, since our algorithms are the first with a sublinear regret bound for both one-shot online learning and continuous bandit problems, there is no previous work to compare with.

**Q4:** Extension to continuous submodular functions. **A4:** Our results only apply to DR-submodular functions. We tried to make it clear in the abstract and throughout the paper. We are sorry for the confusion.

**Response to Reviewer #2:**
**Q1:** Some detailed comments. **A1:** The definition of radius uses the assumption that the constraint contains 0. Assumption 5 is an analogue of the assumption in [23], which assumes that $rB^d \subset \mathcal{K} \subset RB^d$. This assumption is listed in the statement of Theorem 2 (4 *to* 6 on the first line). We assume that $r$ is given.

**Q2:** Applications of RBSM. **A2:** In theory, `RBSM` can be regarded as a relaxation of BSM, which helps us to better understand the nature of BSM. In practice, the responsive model (not only for submodular maximization or bandit) has potentially many applications when a decision cannot be committed, while we can still get the potential outcome of the decision as feedback. For example, suppose that we have an replenishable inventory of $n$ items. Each time there is a customer coming with a utility function unknown to us. We allocate a collection of items to her, and the goal is to maximize the total utility (reward) of all the customers. We may use a partition matroid to model diversity (in terms of category, time, *etc*). In the RBSM model, we cannot allocate the collection of items which violates the constraint to the customer, but we can use it as a questionnaire, and the customer will tell us the potential utility if she received those items. The feedback will help us to make better decisions in the future. Similar examples include portfolio selection: when the investment choice is too risky, *i.e.*, violates the recommended constraint set, we may stop trading and thus get no reward on that trading period, but at the same time observe the potential reward if we invested in that way. We will add more examples in the revision.

**Response to Reviewer #3:**
**Q1:** The resulting regret bounds are worse than the previous one, and shows a limitation of the techniques. **A1:** The previous $O(\sqrt{T})$ bound is achieved by using $\sqrt{T}$ exact gradients or $T^{3/2}$ stochastic gradients per round, while our method only needs one *single* gradient. Our result opens the possibility of achieving sub-linear regret with only one gradient evaluation. We agree that it is an interesting future work to achieve the same $O(\sqrt{T})$ regret bound.

[Meta-Review · NeurIPS 2019]

The reviews as well as the author response were convincing enough (based on discussion) to make a case for this paper.